# Marine spatial planning makes room for offshore aquaculture in crowded coastal waters

S.E. Lester [1], J.M. Stevens[2], R.R. Gentry [3], C.V. Kappel [4], T.W. Bell [5], C.J. Costello[3], S.D. Gaines[3], D.A. Kiefer[6], C.C. Maue [7], J.E. Rensel[8], R.D. Simons [5], L. Washburn[9] & C. White[2]

Marine spatial planning (MSP) seeks to reduce conflicts and environmental impacts, and promote sustainable use of marine ecosystems. Existing MSP approaches have successfully determined how to achieve target levels of ocean area for particular uses while minimizing costs and impacts, but they do not provide a framework that derives analytical solutions in order to co-ordinate siting of multiple uses while balancing the effects of planning on each sector in the system. We develop such a framework for guiding offshore aquaculture (bivalve, finfish, and kelp farming) development in relation to existing sectors and environmental concerns (wild-capture fisheries, viewshed quality, benthic pollution, and disease spread) in California, USA. We identify >250,000 MSP solutions that generate significant seafood supply and billions of dollars in revenue with minimal impacts (often < 1%) on existing sectors and the environment. We filter solutions to identify candidate locations for high-value, low-impact aquaculture development. Finally, we confirm the expectation of substantial value of our framework over conventional planning focused on maximizing individual objectives.

[1] Department of Geography, Florida State University, Tallahassee, FL 32306-2190, USA. [2] Center for Coastal Marine Sciences, 1 Grand Avenue, California Polytechnic State University, San Luis Obispo, CA 93407, USA. [3] Bren School of Environmental Science & Management, 2400 Bren Hall, University of California Santa Barbara, Santa Barbara, CA 93106, USA. [4] National Center for Ecological Analysis and Synthesis, 735 State Street, Suite 300, Santa Barbara, CA 93101, USA. [5] Earth Research Institute, 5843 Ellison Hall, University of California Santa Barbara, Santa Barbara, CA 93106, USA. [6] Department of Biological Sciences, University of Southern California, Los Angeles, CA 90089, USA. [7] School of Earth, Energy & Environmental Sciences, Stanford University, Stanford, CA 93405, USA. [8] Rensel Associates Aquatic Sciences, 4209 234th Street NE, Arlington, WA 98223, USA. [9] Marine Science Institute & Department of Geography, University of California Santa Barbara, Santa Barbara, CA 93106-6150, USA. Correspondence and requests for materials should be addressed to S.E.L. (email: slester@fsu.edu)

This century is marked by the rapid emergence and intensification of human uses of the oceans that present immense economic opportunity, but if not managed properly, could lead to an over-crowded and dysfunctional seascape with serious environmental impacts and costly socio-economic conflicts[1, 2]. Thus, there is a need for ecosystem-based approaches to planning that can strategically and comprehensively balance the location, type, and intensity of ocean user groups, or sectors, across the seascape. Marine spatial planning (MSP) is a place-based, multi-sectoral decision-making approach that is being widely promoted for reducing the conflicts and impacts commonly encountered in conventional sector-by-sector planning[3–6]. In theory, comprehensive and proactive consideration of inter-sectoral interactions and environmental impacts can contribute significantly to the value of MSP over conventional planning[5, 7, 8]. Specifically, conflicts and impacts can be assessed and avoided using an analytical tradeoff analysis that leverages bioeconomic models and explicitly considers sector objectives in the planning decision process[4, 5, 9].

The vast majority of examples of MSP adhere to only some of the attributes of an idealized MSP analytical process[10]. In particular, compared with siting sectors one at a time[5], rarely are multiple sectors sited concurrently, such as coordinated designation of fishery, recreation, aquaculture, and shipping areas[8, 11, 12]. Further, planning is often guided by an implicit consideration of tradeoffs[13, 14], rather than strategically using an analytically defined objective function that considers explicitly the response of each sector in the system. The benefit of an analytical objective function that considers each sector's unique responses to spatial plans is that all stakeholder groups can evaluate a plan's effect on the objectives they value[4, 12]. Some studies have combined different sector responses into a composite metric[15, 16], compromising precision compared with a comprehensive objective function. Others consider some sectors' responses explicitly and others only implicitly (e.g., because their objectives are assumed to be met[12]). Thus, a key gap for MSP science is development and demonstration of an analytical approach for comprehensive, coordinated, and strategic planning—referred to for convenience here as the "full" MSP analytical model—and assessment of its value relative to conventional management.

Planning for offshore aquaculture represents a prime opportunity for MSP[17, 18]. Escalating seafood demand and global seafood trade[19], the near fully exploited state of most wild-capture fisheries, and limited space and resources for expansion of land-based and coastal aquaculture all make offshore aquaculture the next frontier of seafood production[20, 21]. Indeed, nearly all projected growth in seafood production over the coming decades is anticipated to come from aquaculture[22], and offshore aquaculture is a rapidly emerging industry with potential for huge economic and societal benefits[23]. Defined here as occurring beyond the nearshore ( > ~20 m depth), offshore aquaculture comprises multiple sectors cultivating different marine species using various farming technologies[24, 25].

Despite significant potential benefits of offshore aquaculture development, there remain concerns about its environmental impacts and conflicts with other sectors[25], creating social and political opposition to development[26]. For example, in the United States, aquaculture development has been slow in large part because of social opposition and complex and uncertain regulatory and permitting policies[26]. These roadblocks present a significant opportunity for better planning. However, different types of offshore aquaculture produce unique conflicts and impacts—with each other, with other sectors, and with the surrounding environment—that cannot be summarized by a single metric. Further, the location of a farm can have a significant influence on the type and severity of impacts and conflicts with other uses. Thus, optimal siting of offshore aquaculture is a complex MSP problem requiring comprehensive (balancing existing and emerging sector objectives), coordinated (planning multiple emerging sectors simultaneously), and strategic planning (optimized using an analytically defined objective function that explicitly considers the objectives) across the seascape.

We developed, demonstrated, and tested the value of a MSP analytical model that strategically identifies the location, size, and type of offshore aquaculture farms in relation to a suite of existing ocean activities and environmental concerns. We focused on the Southern California Bight, USA (SCB; Fig. 1a), an area with strong interest in and concerns regarding offshore aquaculture development. We constructed spatial bioeconomic models of the productivity and profitability of three representative sectors of aquaculture with industry potential in the SCB: Mediterranean mussel longlines ("mussel"), striped bass pens ("finfish"), and sugar kelp longlines ("kelp"), and applied these models to over 1000 1-km$^2$ planning units (sites) that could possibly be developed for aquaculture (Fig. 1b–d). We also developed models of four key existing "sectors" in the SCB, representing key stakeholder concerns, that could conflict with or be impacted by aquaculture development: the wild-capture California halibut fishery, as halibut use the same soft-bottom habitat that would be developed for aquaculture ("halibut"); the environmental health of the marine benthos that could be degraded by hypoxic conditions caused by excessive organic material released from fish farms ("benthic"); the quality of ocean views from public and private lands that could be blemished by aquaculture surface structures ("viewshed"); and the risk of disease spread among fish farms connected by ocean currents, as disease could compromise the economic viability of aquaculture and the health of the SCB ecosystem ("disease"). We then integrated the 7-sector meta-model with an analytical tradeoff analysis to derive optimal spatial plans for the development of mussel, finfish, and kelp aquaculture that simultaneously minimize inter-sectoral impacts and maximize individual sector values. In the optimization, we considered a range of sector-specific weighting factors to reflect alternative societal preferences and/or levels of political influence for how much and what types of aquaculture development are desirable and what degrees of impacts are acceptable.

We identify thousands of optimal spatial plans, and map a small subset of those plans that could be especially informative for decision-making. Optimal plans have minimal impacts to a wild fishery, viewshed quality, and the health of the benthic environment, and minimize the risk of disease outbreaks, while generating significant revenue and seafood supply from marine aquaculture development. We find that by using our model, sector values can be substantially increased (by millions of dollars) and impacts can be reduced (to < 1%), compared to using conventional approaches to spatial planning. More generally, our rigorous and flexible framework can minimize tradeoffs arising from the inevitable expansion and intensification of a wide variety of human uses of the oceans.

## Results

**Sector tradeoffs.** Solving the objective function for all combinations of the seven sectors, each with one of six weighting factors (ranging from low to high priority for maximizing/minimizing the value or impact of the sector), we identified $6^7 = 279,936$ optimal spatial plans (i.e., exact analytical solutions given the sector values and sector weights specified in the objective function). Collectively, the plans delineate a 7-dimensional "efficiency frontier" of optimal MSP solutions (Fig. 2a). Each solution represents a SCB-wide aquaculture development plan (location and type—mussel, finfish, kelp or none—across 1061 1-km$^2$ potentially developable sites) that best minimizes sector impacts and maximizes sector values to the extent possible and relative to their level of socio-political preference (applied as weights).

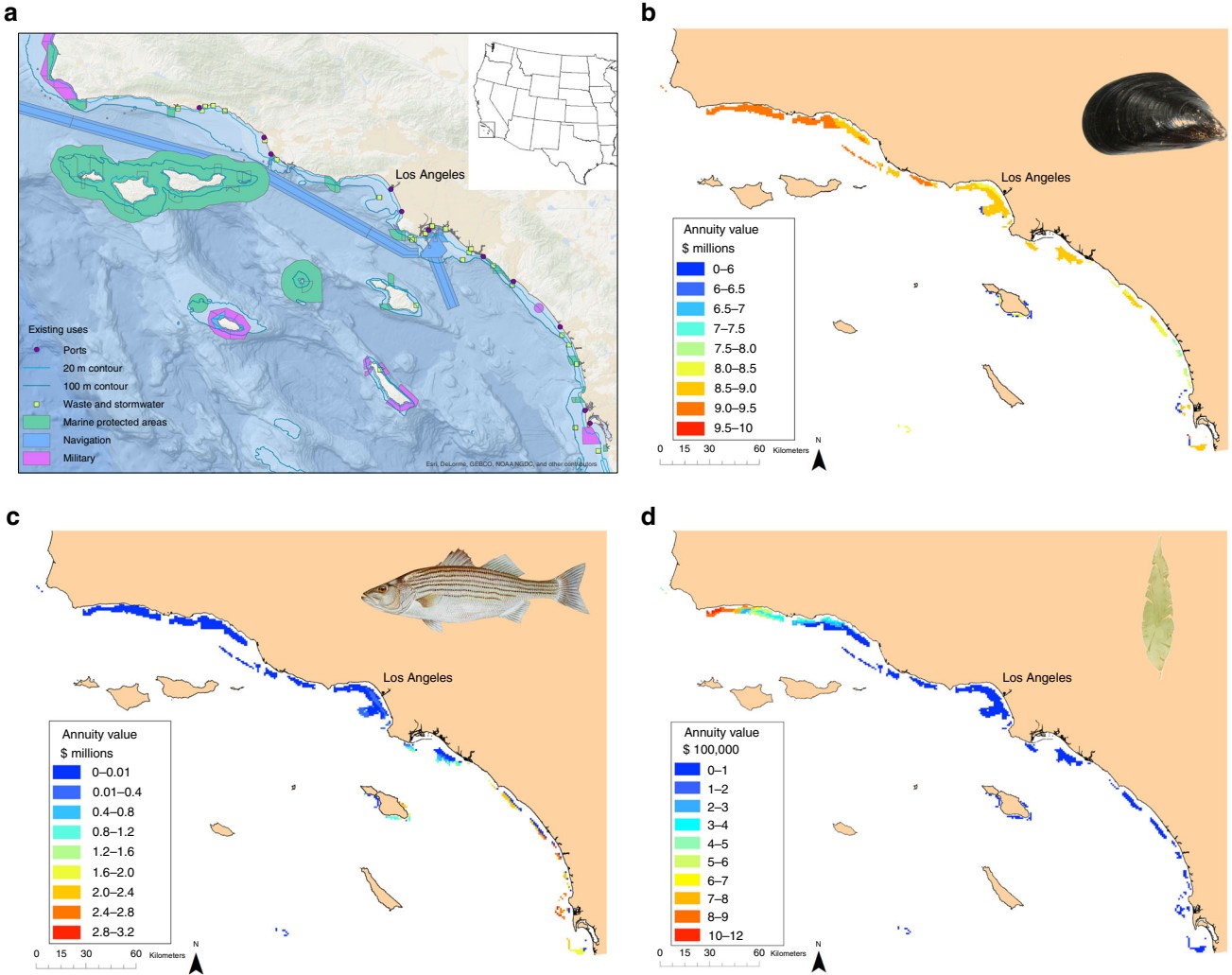

**Fig. 1** Study domain, spatial constraints, and potential value for aquaculture development. **a** Select regulatory and logistical constraints to aquaculture development in the SCB. Potential annuity ($/year) in each developable site for (**b**) mussel, (**c**) finfish, and (**d**) kelp aquaculture sectors. Mussel image in **b** from https://commons.wikimedia.org/wiki/File:HPIM1476a.jpg

Although aquaculture development in the SCB could cause considerable conflicts and impacts, explicit mediation of this problem with tradeoff analysis reveals that such outcomes need not be severe, particularly when aquaculture is restricted in its levels of development. For example, unrestricted development of mussel farms could reduce halibut fishery value by ~7%, a relatively low percentage that nonetheless represents >$100,000 in lost annuities (equivalent annual annuity of net present value (NPV)) to that sector. In contrast, we found strategic development of up to 25% of the maximum value of mussel aquaculture to reduce the value of the halibut fishery by a mere 0.2%, or just ~$3700 in lost annuities, while generating ~$2 billion in annuities to the mussel aquaculture industry. Further, the most profitable sites for kelp and finfish aquaculture are concentrated away from the halibut fishery's most valuable areas, and thus under MSP those aquaculture sectors can be nearly fully developed with virtually no impact on the fishery. MSP also results in minimal viewshed impacts (<1% reduction in value) when the three aquaculture sectors are limited to <25% of their full development value and they avoid key locations near populated coastal areas. Similarly, disease risk is concentrated in specific areas due to ocean currents generating high levels of connectivity among certain sites. As a result, MSP can suppress the risk of disease spread by avoiding siting finfish farms in highly connected sites, while still generating up to 88% of potential maximum finfish aquaculture value.

MSP can also mediate competition between aquaculture sectors. Mussel farming is profitable throughout our study region and if developed first could preclude nearly all finfish and kelp aquaculture development (Fig. 1b). Yet optimal spatial plans allow for profitable development of all three aquaculture types; for example, finfish and kelp aquaculture can achieve nearly their full potential value concurrent with mussel aquaculture achieving ~50% of its potential value. This outcome is not generated from simply developing mussel aquaculture where it does not conflict with the other aquaculture sectors, but rather from coordinated, strategic planning that considers the relative value of each site to all sectors. Strategic planning does not, however, guarantee avoidance of all tradeoffs. For example, the risk to benthic environmental health increases nearly linearly with each additional site developed for finfish aquaculture, because environmental effects from finfish farms are relatively consistent across the planning domain. On the other hand, in some cases (e.g., finfish and kelp aquaculture) there is no interaction, and thus no tradeoff between sectors, enabling each to potentially achieve its maximum value.

**Development hotspots**. Visualization of the frequency of development of each type of aquaculture in each site across all of the MSP solutions reveals locations that are generally favorable to

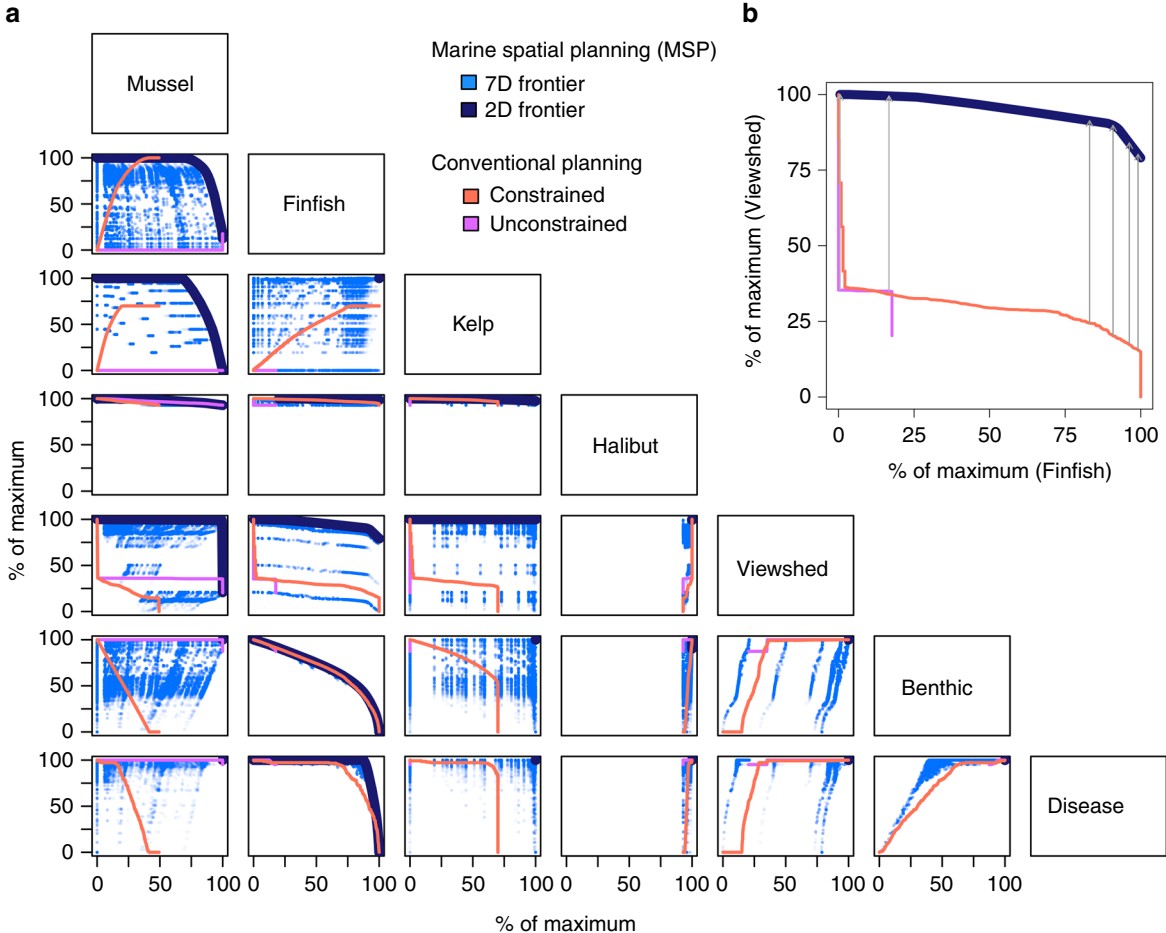

**Fig. 2** Marine spatial planning and conventional planning outcomes. **a** Pairwise (2-D) and 7-D efficiency frontiers representing 279,936 optimal spatial aquaculture development plans determined by the MSP objective function, as well as outcomes from spatial plans expected under conventional planning (see legend). **b** Example with arrows showing the value of MSP over conventional planning

aquaculture development regardless of socio-political preferences for particular sectors (Fig. 3). Note, the visualization is not of just a set of estimated solutions to one parameterization of the MSP problem (e.g., as often done using Marxan with Zones[12]), but of all the optimal solutions derived for each of the weighting factor scenarios considered. Thus, the "hotpots" in Fig. 3 could guide a MSP process in the SCB by highlighting specific sites that will be more appropriate for development regardless of socio-political preference, providing a more tractable planning tool in cases where examining many possible optimal plans and/or precisely specifying the weighting preferences for all objectives is not feasible. Comparison of these hotspot maps (Fig. 3b–d) with the distributions of the aquaculture sectors' potential value across the SCB (Fig. 1b–d) reveals that MSP generates a substantial departure in development plans from those expected by single-sector planning focused solely on aquaculture profit. For example, under MSP the most consistently developed mussel sites are largely clustered in the central portion of the SCB (Fig. 3b), despite the most profitable sites for the sector being located in the north. Further, aquaculture development is minimized in the southern SCB where there are high-value halibut fishing grounds and where viewshed impacts would be highest.

**Hypothetical planning exercise.** To highlight the utility of MSP for identifying a set of spatial plans that meet specific policy objectives, we filtered the 279,936 MSP solutions to those plans for which the impact of aquaculture development on each of the

existing sectors is no >5% of their value, while each of the aquaculture sectors must achieve at least 5% of their value. This procedure yielded 450 spatial plans (Fig. 4a). Despite the strict impact constraints, nearly a fifth of the developable sites are developed in this filtered set of spatial plans. Kelp farming, on average, achieves the highest relative value among the aquaculture sectors due to its relatively low impacts, and finfish achieves the lowest relative value, because it impacts all four existing sectors. These filtered results could inform regulators on how much development to allow (Fig. 4a), and where to develop (Fig. 4b) each type of aquaculture in order to meet a given policy specifying acceptable impacts.

Useful for a negotiation process is the ability for managers and stakeholders to compare a small number of distinct MSP solutions, or seed plans, that all generate acceptable outcomes. Accordingly, we used cluster analysis to identify five seed plans that represent the maximum amount of variation in spatial design among the 450 filtered plans (Fig. 5a). Although these plans specify different locations for development of the three types of aquaculture (Fig. 5b–d), they all achieve considerable aquaculture value ($589 million–2 billion, $33–51 million, and $80–$181 million in annuities to the mussel, finfish, and kelp sectors, respectively) while minimizing impacts to the existing sectors (0–5% impact). Even if modified by stakeholders, these plans are likely to produce near optimal outcomes[9].

**Value of marine spatial planning.** To assess the value added by our MSP approach, we compared solutions along the efficiency

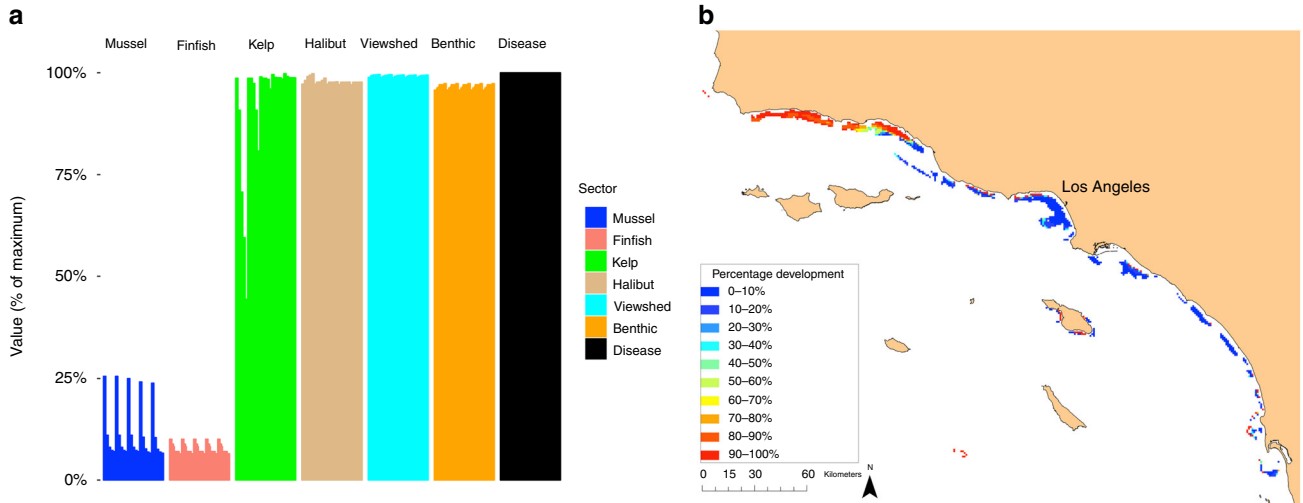

**Fig. 3** Hotspot maps of potential development. The percentage of the 279,936 optimal spatial plans containing each site in its developed state for (**a**) any form of aquaculture, and (**b**) mussel, (**c**) finfish, and (**d**) kelp aquaculture

**Fig. 4** Hypothetical planning exercise. **a** Subset of optimal spatial plans in which each aquaculture sector achieves > 5% of its maximum possible value, and no existing sectors are impacted by > 5%, resulting in 450 plans. **b** Given these 450 plans, percentage of plans in which each site was developed for aquaculture

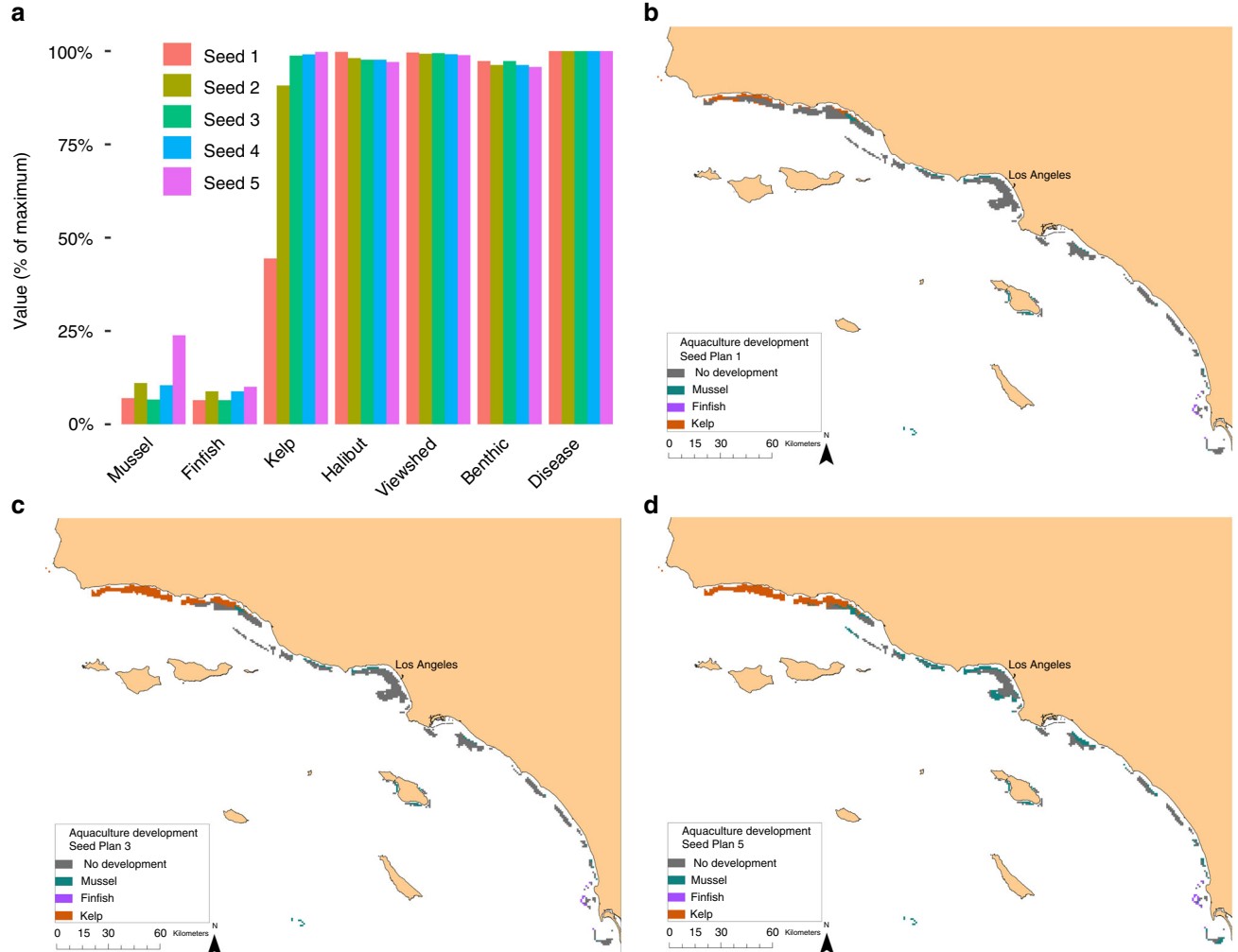

**Fig. 5** Seed plans selected from the filtered set using cluster analysis to represent maximum variation in spatial planning design. **a** Resulting value of each sector in each seed plan. **b–d** Maps of three of the seed plans as indicated in the legend of panel **a**

frontier with outcomes expected under modeled representations of conventional planning. We assumed that conventional planning considers individual values and cumulative impacts of the aquaculture sectors but is neither comprehensive (considering individual impacts) nor coordinated (via simultaneous planning, Fig. 2b). We considered two possible characterizations of conventional planning: unconstrained aquaculture development, and constrained development that drives a balanced footprint of mussel, finfish, and kelp farms. For both conventional planning approaches, we found that every sector does as well or better with MSP (Fig. 6). For the four existing sectors, the benefits from MSP typically range 0–100% and increase with the level of aquaculture development. For the aquaculture sectors, the benefits from MSP also range 0–100%, but typically decline with aquaculture development.

The value of MSP is sensitive to the type of conventional planning examined (Fig. 6). MSP benefits finfish and kelp aquaculture little relative to constrained planning, but substantially (often doubling their values) relative to unconstrained planning that allows the mussel sector to dominate aquaculture development because of its superior value/impact ratio at most sites. For existing sectors, constrained conventional planning performs similarly to MSP at low levels of aquaculture development, because regulating for an equivalent footprint among the three aquaculture sectors restricts mussel

development, thereby limiting impacts initially. However, the efficiency of this approach compared with MSP deteriorates at higher levels of aquaculture development because kelp and finfish sectors are allowed to develop low value, high impact sites, relegating mussel development to lower mussel value sites.

## Discussion

MSP is widely acclaimed as an essential tool for reducing conflicts among management objectives[7]. But current scientific frameworks and applications rarely achieve the trifecta of comprehensive (balancing existing and emerging sector objectives), coordinated (planning multiple emerging sectors simultaneously), and strategic (optimized using an analytically defined objective function that explicitly considers the objectives) planning. We developed a generalizable approach to execute a "full" MSP analytical model constructed to meet these objectives and demonstrate its utility when applied to the challenge of offshore aquaculture development in California. Offshore aquaculture in California, as in many other regions, is being met with some opposition from environmental regulators, coastal residents, and commercial ocean users[25, 27, 28]. However, as an emerging use that could contribute to economic development and sustainable seafood production, aquaculture offers a ripe opportunity for proactive planning. By modeling different types of offshore

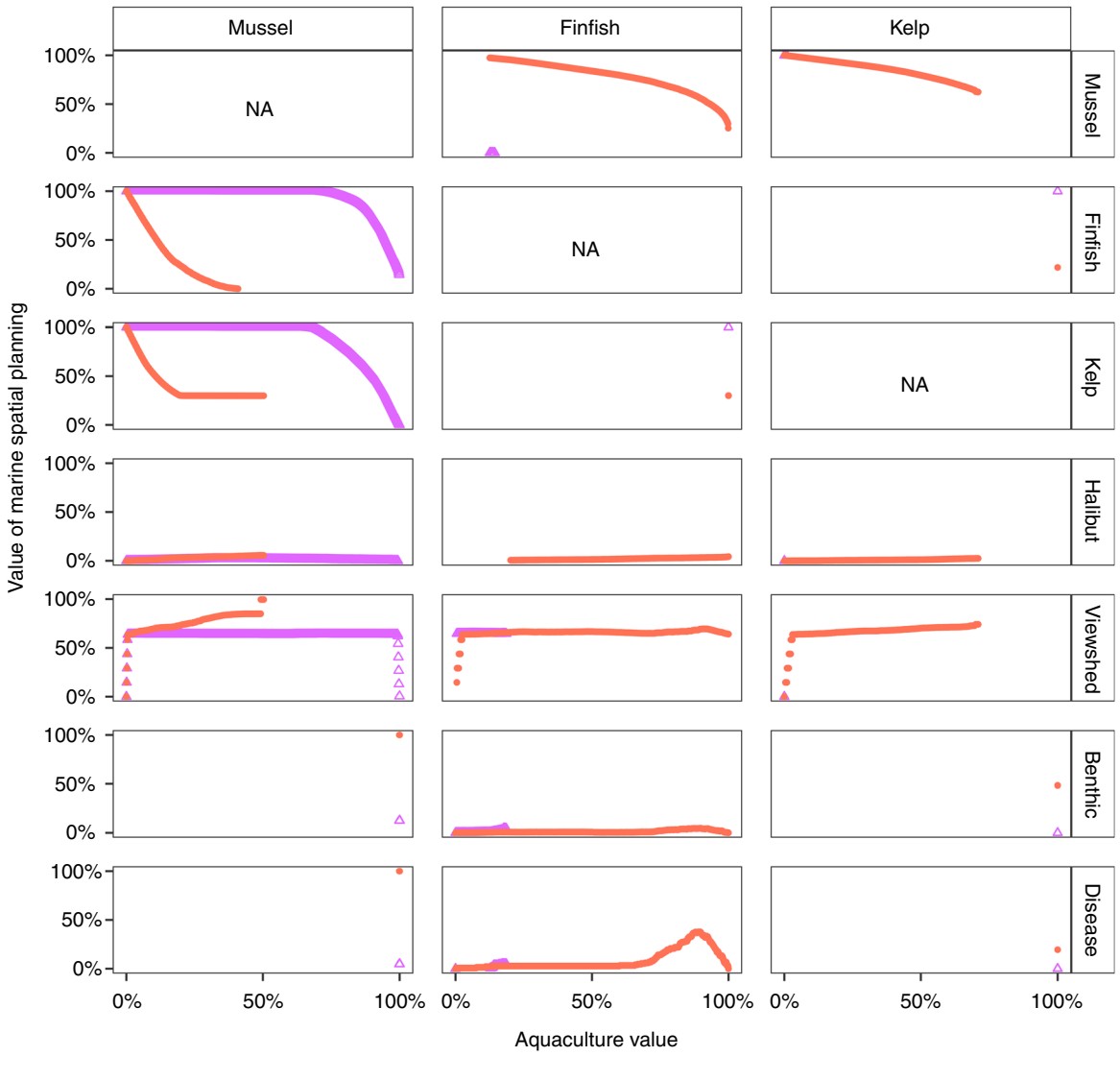

**Fig. 6** Value of MSP in relation to constrained (red filled circles) and unconstrained (pink open triangles) conventional planning, and for each sector in relation to level of development of each aquaculture sector

aquaculture and its potential impacts on different ocean uses and the environment, and integrating multiple objectives into a tradeoff analysis, we provide a method for spatial planning that is comprehensive, coordinated, and strategic.

Our analytical approach and case study results yield several important insights. First, dramatic tradeoffs were unavoidable only at high levels of aquaculture development. Using MSP, it is relatively easy to minimize and in some cases eliminate tradeoffs when aquaculture is kept at moderate levels of development. Further, these moderate levels still generate substantial economic returns for the aquaculture industry. For informational purposes, we looked across the full range of potential aquaculture development (0–100%), but in practice the starting point for the SCB (i.e., its current state) is almost no offshore aquaculture, and regulatory constraints restrict substantial development[28]. In the coming decades, the region is likely to experience only modest development (e.g., < 50 km$^2$, equating to <30% maximum total potential value per aquaculture sector in our model). Consequently, the region of the tradeoff curves that is most relevant for guiding strategic aquaculture development in southern California suggests that MSP could greatly improve aquaculture without

significant negative consequences for existing uses or the environment.

Conducting MSP obviously will not eliminate stakeholder opposition to development, or its impacts. Given widespread negative perceptions of aquaculture that have been reinforced by the media and various interest groups[26], there is the need for public education about the actual impacts and benefits expected from aquaculture development when guided by scientifically informed planning, as described by this study. Additionally, our model includes only some of the potential impacts of aquaculture development; there may be other "costs" that we did not consider (e.g., genetic transfer from farmed to wild stocks, reducing the latter's fitness[29]) that could influence our results and stakeholder perceptions of aquaculture. Importantly, previous work on spatial planning for offshore aquaculture has focused primarily on identifying suitable sites in terms of technological and biophysical constraints for the farm and farmed species or accounting for potential impacts[30–33]; exploring interactions and synergies between offshore aquaculture development and other economic uses of the ocean (e.g., wind farms, oil platforms, nearshore aquaculture[34–36]); or examining tradeoffs with another single

objective[37]. However, none of this work has used a multi-objective tradeoff analysis and optimization to inform siting decisions. Our finding of potential compatibility between aquaculture development and a suite of existing ocean uses and management priorities, thus, represents an advance of this existing literature.

We also demonstrate that values of the seven sectors under MSP typically exceed and are never lower than their values under conventional planning. This increase in value from MSP is sometimes as high as 100%, even under low levels of aquaculture development (e.g., viewshed impact can be nearly entirely eliminated when finfish development is limited). These relative values translate into significant absolute benefits, for example, up to millions of dollars in annuities to aquaculture sectors and tens of thousands of dollars in avoided lost annuities to the halibut fishery. The precise value of MSP varies depending on the reference form of conventional planning, but, importantly, the basic qualitative result that MSP provides higher value is insensitive to the exact definition of conventional planning. MSP's value to existing sectors increases with the level of aquaculture development because, at high levels, greater potential impacts from poor planning are at stake and can be avoided using MSP. This result highlights the value in using MSP to design a comprehensive master plan guiding future potential development rather than using strategic models to site individual farms one at a time. Furthermore, as the level of development of one type of aquaculture increases, the value of MSP to the other aquaculture sectors declines, because MSP restricts their development in order to protect existing sectors from impact. Thus, in industries like aquaculture or offshore energy with multiple emerging sectors, sectors lagging in development would benefit from promoting MSP of the entire industry from the start.

The value of MSP is positive because, unlike conventional planning, MSP explicitly considers sector tradeoffs, seeking solutions that efficiently maximize value and minimize impacts[4, 5, 38]. This value can provide a powerful incentive to adopt an improved analytical model for guiding MSP. This incentive is aligned for new industries, who are likely to face less opposition while still gaining access to valuable locations; for existing sectors, who recognize the inevitability of new development and seek to minimize the negative impacts they experience; and for planners and policy-makers, who are interested in efficient outcomes that make society better off. While the likelihood and pace of new uses in the oceans is variable around the world, there is a general trend towards new development including aquaculture, renewable energy, oil and gas extraction, and seabed mining[39]. In parallel, there is widespread concern that traditional approaches to marine management have failed to protect the environment and existing uses[2, 40]. With this comes considerable momentum in many places for MSP and for improved analytical frameworks to guide this planning[14, 41, 42], and often the locations leading the way are those facing impending new uses[5, 43]. Our comprehensive approach to spatial planning provides a platform for delineating and allocating rights and responsibilities among the various stakeholders affected by the planning process.

This study generates a methodological advance over previous research that has applied spatial bioeconomic models and tradeoff analysis to MSP[9, 10, 12, 44]. Similar to our study, Yates et al.[15] coordinated development of multiple emerging ocean sectors (wind and tidal energy); however, unique responses by different existing sectors (fisheries) were not considered explicitly in their objective function, limiting their ability to derive optimal solutions. Metcalfe et al.[16] similarly combined their analysis of sector responses (among independent fisheries, in their case), limiting the optimality of their solutions. White et al.[5] did consider unique sector responses in their MSP objective function; however, their analysis considered development of only one emerging sector

(wind energy) within a much smaller planning domain (84 sites). As a result, they were able to use a heuristic to estimate (but not necessarily identify) the MSP efficiency frontier. Expanding the analysis to consider multiple emerging and existing sectors and several management options per site within a larger planning domain (1061 sites) complicates the problem substantially (i.e., the discrete choice space becomes unwieldy). We solved this issue by developing an optimization algorithm that quantifies sector responses to each possible planning decision on a site-by-site basis and identifies spatial plans in relation to the sum of these weighted values. By explicitly comparing alternative spatial plans in this way, the complexity of the planning problem is reduced, and reliance on a heuristic eliminated (i.e., exact solutions are derived). The computational time saved enables the framework to be applied to alternative sector weighting scenarios, not just alternative estimates of solutions to a few unique parameterizations of the framework[11], in order to assess the potential range of tradeoffs among the sectors in the system. Further, because our framework evaluates all sector objectives explicitly, we are able to provide details about the tradeoffs specific to each sector-by-sector interaction. But perhaps most notably, models of additional sectors can be integrated into the framework, making it flexible for informing real-world MSP processes.

A downside of our approach is that it requires static models. Accordingly, to account for sectors expected to react dynamically and uniquely across sites (e.g., the halibut fishery, via larval spillover, adult movement, and redistribution of fishing effort), we used static models for identifying optimal plans, but then used dynamic models to simulate the actual outcome of each MSP and conventional planning solution for the halibut fishery (similar to the approach used by[16]). This two-step approach enabled us to estimate the actual, dynamic implications to the alternative spatial plans, in contrast with previous analyses based entirely on static models, e.g., Marxan with Zones[12, 15].

As with any attempt to model numerous ocean uses and their complex interactions, we made simplifying assumptions that may affect our results. For example, we assumed best-management practices within each aquaculture farm, including low farm densities within each site. We also assumed sufficient global seafood demand to maintain constant prices[21]; such an assumption could fail at high levels of aquaculture development, though probably not for a region the size of the SCB[21]. For existing sectors, our estimates of impacts were necessarily based on indirect and/or incomplete metrics due to data and model limitations. For example, our proxy for ecosystem health was flux of organic material from fish farms to the benthos, which at high levels can generate hypoxic conditions hazardous to marine organisms (but at low levels might actually be beneficial for the ecosystem). For disease, we focused on viruses, because they are difficult to contain with best-management practices (e.g., with antibiotics) and present a high risk for propagating new diseases[45], but we acknowledge there are other important types of diseases. Finally, we focused on three representative types of aquaculture and four existing sectors of high concern. However, there are other classes of aquaculture not captured by our study (e.g., integrated multi-trophic aquaculture), as well as other sectors that could be impacted by aquaculture (e.g., marine mammal and seabird conservation; shipping and navigation) or co-located with it, e.g., ocean wind farms[34]. If important sectors are missing from our analysis, then the spatial planning solutions will not necessarily be optimal (though they may be close[9]). However, with high-quality spatial data and a detailed understanding of their interactions, these other sectors could be integrated into our MSP framework.

There is considerable evidence that MSP is more likely to succeed when there is a participatory process with adequate

stakeholder engagement[46–48], thus it is important that our analytical approach can operate within the realities of socio-political planning processes. Ideally, scientists and managers would discuss and validate model outputs with stakeholders and adjust the models accordingly[9]. Practical and local knowledge from stakeholders can also be integrated with scientific knowledge, including within a tradeoff analysis, to produce more effective environmental policy[49–51]. This information could subsequently inform the delineation of rights and responsibilities to various stakeholder groups, using the spatial planning outputs as an overall framework, much like would occur in a zoning process on land with property rights and responsibilities within each zone. A potential hurdle to using our approach in a participatory process is that the vast number of optimal plans identified may be daunting for stakeholders and planners. However, we have shown ways to distill such a large set of optimal results into simple guidelines for informing planning (hotspot maps), as well as how to select from the numerous optimal results a subset that is more manageable to review (filtered and seed plans from the hypothetical planning exercise). Finally, an important next step is for technical modeling approaches like the one presented here to be translated into user-guides for managers and policy-makers so that cutting-edge analytical approaches can inform practical decision-making.

We provide an analytical advance over previous MSP approaches that allows comprehensive, coordinated, and strategic planning for multiple emerging and existing sectors. We applied our framework to offshore aquaculture, which is expected to grow rapidly worldwide and is likely to become an essential component of future food production. Our case study is set in southern California, a crowded coastal marine ecosystem with diverse management objectives representative of many highly populated coastlines around the world. There and elsewhere, strong interest in offshore aquaculture development by government and industry is rivaled by diverse opposition driven by perceived spatial conflicts with existing resource users and environmental impacts[25]. The methods and findings from our study may temper opposition because they show that carefully planned aquaculture can generate high-value while ameliorating negative effects to existing sectors. We also demonstrate that conventional planning, even while including environmental regulations and moderate coordination among sectors, remains less effective compared with our MSP approach—a result that corroborates previous studies on marine spatial planning[5, 8]. Although we focus on aquaculture development in the SCB, our models can be adapted to aquaculture development elsewhere (including outside of the United States), to sectors beyond aquaculture, and to spatial planning in areas where there are opportunities for simultaneously managing existing and emerging ocean sectors, provided there are sufficient data[52]. Our approach opens up new opportunities for improved ocean management to achieve sustainable and productive use of marine resources.

## Methods

**Study domain**. See Supplementary Table 1 for an overview of the steps in our MSP analytical model. We first defined our case study domain to encompass the Southern California Bight (SCB), stretching from Point Conception in northern Santa Barbara County, California to the US-Mexico international border south of San Diego, California. We divided the area into a 1-km² resolution planning grid (ESRI ArcGIS 10.2 Tool: Create Fishnet), resulting in a model domain containing 6425 1-km² planning units or "sites." All geospatial data layers used the California Teale Albers NAD83 projection.

**Fixed constraints on offshore aquaculture development**. Fixed constraints were applied to determine whether a given 1-km² "site" was potentially developable for each of the three types of aquaculture. First, based on current practice in the offshore aquaculture industry, we applied a depth constraint of 20–80 m for mussel and kelp aquaculture and 30–100 m for finfish aquaculture. Minimum and

maximum depths in each site were calculated in ArcGIS using the Southern California Coastal Relief Model (Supplementary Data 1; ArcGIS 10.2 Tool: Zonal Statistics as Table). A site was considered potentially developable for a given aquaculture type if its minimum and maximum depths fit within the aquaculture type's depth constraint.

Second, to identify locations with appropriate bottom type for aquaculture development, and to quantify habitat availability for use in the wild-capture halibut fishery model, we mapped soft and hard bottom substrate across the SCB using best available data. In state waters (0–3 nm from shore), the most comprehensive dataset for this purpose was compiled for the California Marine Life Protection Act process and made publicly available by the California Department of Fish and Wildlife (CDFW) (Supplementary Data 1). To fill gaps in the CDFW data and extend the dataset beyond state waters, we used the California Geology Series statewide continental margin habitat layer, downloaded from the Seafloor Mapping Lab of California State University, Monterey Bay (SFML) (Supplementary Data 1). Both data layers classify seafloor into soft sediment (sand and mud), hard bottom (rock, boulders, and gravel), mixed hard and soft, or unknown. We merged these two datasets to make a comprehensive data layer for the SCB. In order to fill a remaining gap in seafloor habitat data in the San Diego area (i.e., an area classified as "unknown"), where there has been particular interest in finfish aquaculture development, we classified higher resolution (10 m) bathymetry data, collected by the California Seafloor Mapping Program but at the time not yet publicly released, into hard and soft bottom (Supplementary Data 1). Hillshade and Vector Ruggedness Measure (VRM) layers were derived from the bathymetry and provided for our use by L. Wedding (personal communication, Stanford University, unpublished data; Supplementary Data 1). Based on the range of thresholds used previously by SFML to classify similar data e.g., in ref. [53], we applied a rugosity threshold of 0.0004 to the VRM layer as the classification cutoff between hard vs. soft bottom. This specific threshold produced the best classification, based on visual comparison to the hillshade layer. We then used the classified data to fill in unknown habitat in the area off of La Jolla, CA. We were unable to obtain further data to fill other gaps in the substrate data layer (2725 1-km² sites were classified as "unknown").

Existing offshore aquaculture permit holders and applicants in California are required to demonstrate that their farms are not proximate to or above hard bottom habitat and will not negatively impact such habitat[54, 55]. For this reason, we assumed that each site must have 100% soft bottom habitat to be considered developable. Hard bottom and mixed soft and hard bottom could not be developed, and we also conservatively prohibited unknown areas from development. Future habitat mapping could fill unknown habitat gaps and potentially expand the developable area for aquaculture in the SCB.

Third, we assumed aquaculture would be restricted from sites with certain human uses or impacts. All uses and designations that we assumed would preclude aquaculture development were mapped and overlaid with the planning grid, and intersecting grid sites were classified as undevelopable. Specifically, we eliminated marine protected areas (MPAs) that prohibit aquaculture and/or any modification of the seafloor, including the Channel Islands National Marine Sanctuary (CINMS) (Supplementary Data 1; Fig. 1a). Although the Marine Protection, Research, and Sanctuaries Act of 1972 that defines policy for CINMS does not explicitly prohibit aquaculture, it does prohibit modifications to the seafloor[56]. We interpreted this as indicating that mooring of buoys and lines for aquaculture within CINMS would not be allowed. We also excluded existing offshore energy infrastructure (i.e., oil rigs), military zones, anchorages, and Traffic Separation Scheme shipping lanes, all mapped using the de facto MPA data layer provided by the National Marine Protected Areas Center (Supplementary Data 1; Fig. 1a). Lastly, we limited aquaculture development from the vicinity of offshore, subsurface treated wastewater effluent outfalls and major river mouths (Supplementary Data 1; Fig. 1a), in accordance with current water quality regulations[28], by excluding sites containing outfalls or river mouths, as well as the eight immediately adjacent sites, based on estimates from the region of the average dilution and dispersal distances of wastewater and stormwater runoff plumes[57–61]. Lastly, for each aquaculture type, sites with negative value (NPV and annuity) were assumed to be undevelopable.

After accounting for all of these regulatory, logistical, and economic constraints to aquaculture development, 1061 sites were determined to be potentially developable for one or more types of aquaculture (Supplementary Fig. 1). We recognize that some of the constraints we included might be more flexible than our analysis assumed (e.g., aquaculture could be co-sited with decommissioned oil rigs; CINMS could allow aquaculture moorings) and that conversely, we might be overlooking some fixed constraints (e.g., classified military zones).

**Bioeconomic models**. We constructed spatially explicit models to estimate the distribution of value of three emerging aquaculture sectors and four existing sectors impacted by aquaculture development. For each aquaculture sector we developed a bioeconomic model estimating annual yield and profit (Supplementary Fig. 2) for all developable sites given a specified fixed farm design for each type of aquaculture. We also took into account environmental conditions (e.g., water temperature, surface currents, particulate organic carbon levels, nitrate concentrations, photosynthetically active radiation, depth, distance to port, and wave height) in each site and accounted for start-up costs followed by fixed farm operational costs specific to

each type of aquaculture (Supplementary Fig. 3), assuming static market conditions. Mussel and kelp aquaculture models were modified from published individual growth models and scaled up to the farm level[62, 63]. Finfish aquaculture was modeled using the aquaculture siting, production and impacts model, Aqua-Model[64]. For each aquaculture type in each site, we amortized annual profits from each site, $\pi_t^i$, in relation to an economic discount rate, $\delta$ (5% in this case), and then summed the discounted profits to estimate the NPV to the sector over a 10-year time horizon[65, 66].

$$\mathrm{NPV}^i = \sum_{t=0}^{T=10} \frac{\pi_t^i}{(1+\delta)^t} \quad (1)$$

We also estimated the equivalent annual annuity of each sector's NPV, which represents the series of even cash flows received by a given sector over the 10-year time horizon for site $i$. Note that annuity is simply NPV multiplied by a constant.

$$C^i = \frac{\delta(\mathrm{NPV}^i)}{1-(1+\delta)^{-T}} \quad (2)$$

A 10-year time horizon (starting with the present year) was chosen due to the rapid projected growth and innovation in the aquaculture industry, which is expected to generate high turnover of aquaculture technology on a decadal scale[67]. This timeframe also matches the permit renewal cycle specified for aquaculture in US Federal waters of the Gulf of Mexico according to the recently implemented offshore aquaculture Fishery Management Plan[68].

Sites without positive profits for an aquaculture type were set as undevelopable for that type. These economic constraints, in addition to the regulatory and logistical constraints, resulted in 1011 potentially developable sites for mussels, 329 for finfish, and 325 for kelp. Admittedly, as a nascent industry, aquaculture technology and expertize is likely to show considerable improvements in the future, which could lower costs and increase production and revenue, resulting in a larger number of developable sites, but we focused this analysis on current economic and technological conditions. We considered the full range of development across the planning domain (i.e., development of 0–100% of profitable sites for each aquaculture type).

We constructed models to estimate aquaculture impact on four existing "sectors" that represent common social concerns with offshore aquaculture development.

Wild-capture fisheries: We modeled spatial and temporal population dynamics of California halibut, *Paralichthys californicus*, including larval dispersal based on a Regional Ocean Modeling System (ROMS), and adult movement in relation to inter-site distance and habitat quality (Supplementary Fig. 4). We assumed sites with aquaculture development would be closed to halibut fishing (e.g., due to risks of gear entanglement) and estimated resulting changes in fishery yields and profits from commercial and recreational harvest (Supplementary Fig. 5). The equilibrium output of this dynamic model was used for determining MSP and conventional spatial plans, but then we estimated the value of the fishery in relation to the plans using the fully dynamic model.

Viewshed quality: We estimated visual impacts as the number of coastal residents and visitors who could see a farm in each developed site, using a cumulative viewshed model in ArcGIS with coastal population density, visitation rates to state parks and beaches, coastal elevation, and distance to the farm as input data, and assuming that mussel and kelp farms are visible within a 3-km radius and finfish farms are visible within an 8-km radius (Supplementary Fig. 6a, b).

Benthic environmental health: We used AquaModel[64] to estimate the spatial distribution and rate of total organic carbon flux from fish farms to the seafloor as a proxy for effects on the benthic ecological community (Supplementary Fig. 6c). Although low levels of organic material could be harmless or even beneficial to the benthic community, we conservatively assumed that higher levels of flux correspond to increased benthic impact, as higher delivery rates of organic material increase the risk of hypoxic conditions.

Disease risk: We focused on assessing viral disease risk, because bacterial disease has been relatively well controlled through vaccines and antibiotics in modern finfish aquaculture. To assess the spatial planning dimension to viral disease risk, we assessed the oceanographic connectivity of viral particles among finfish farms because farm location and density in relation to ocean currents could influence the risk of disease transmission and system-wide outbreak. We estimated the relative risk of virus spread for finfish farms as the sum of the eigenvector centrality indices of the developed sites, calculated using a connectivity matrix derived for all developable sites using a ROMS parameterized for marine virus life history (Supplementary Fig. 6d). Mussel and kelp farms have received less attention for their potential disease risk and are unlikely to pose a threat to the benthos in the offshore environment, so were assumed to have no impact on these sectors. See the Supplementary Notes for a detailed description of each of the models mentioned above, Supplementary Data 1 for spatial data layers used in our analyses, and Supplementary Data 2 for model parameter values and supporting references.

**Tradeoff analysis**. We developed a spatial selection model that considers interactions among the seven sectors and optimizes management decisions based on weighted socioeconomic priorities for each of the sectors in maximizing their gains or minimizing their impacts. The tradeoff analysis contains the above-described seven sector models (mussel, finfish, and kelp aquaculture, and halibut fishery, viewshed, benthic impact, and disease risk), which are used to estimate spatially explicit potential values of each sector in response to each form of development at each site in the study domain. These seven models are combined by the tradeoff analysis into a meta-model that is solved in relation to a MSP objective function. The objective function considers the potential value of each site by each sector in relation to four separate development options (no development, mussel development, finfish development, kelp development). These values are weighted and then summed across sectors to calculate an aggregate metric at a site for each development option. For a given set of sector weights, the goal of the tradeoff analysis is to determine what development option should be chosen at each site in order to maximize the aggregate metric in the MSP objective function. The solutions generated for all weighting scenarios represent the set of optimal MSP development plans for the three aquaculture types in the SCB.

Currently, no offshore aquaculture exists in the SCB, and planning procedures for determining the number and location of suitable sites, and the level of acceptable impacts on existing sectors, are in the preliminary stages[28]. To simulate the full range of possible relative socioeconomic priorities across the sectors (aquaculture and existing), we assigned priority weights to each of the seven sectors. Each sector's weight ranged from 0 to 100% in increments of 20% (i.e., $\alpha^n = 0, 0.2, 0.4,\ldots1$, for each sector $n$). Evaluation of all sector-by-sector combinations generates $6^7 = 279{,}936$ weighting combinations among the seven sectors. Each combination is then evaluated within the tradeoff model, whereby we optimize each unique objective function to identify optimal spatial plans.

Let $V_{n,i,p}$ be the value to sector $n$ at site $i$ from pursuing development option $p$ at that site. There are seven sectors ($n = \{$mussel, finfish, and kelp aquaculture; halibut fishery; viewshed; benthic health; and disease risk$\}$), 1061 sites ($i = \{1,\ldots,1061\}$), and four development options ($p = \{$develop mussel aquaculture, develop finfish aquaculture, develop kelp aquaculture, and no aquaculture development$\}$). $V_{n,i,p}$ depends on the "response" of a sector, at a site, to a particular development option. For each aquaculture sector, the response is the equivalent annual annuity (in dollars) generated to the sector if the site were to be developed for that type of aquaculture. For the halibut sector, the response is the annuity (in fishery yield) to the halibut fishery at the site if it is not developed for aquaculture, and otherwise zero if aquaculture is developed there because the halibut fishery is excluded from the site. For viewshed, the response is no impact (i.e., no reduction in number of person views of a site) if there is no development there, partial impact if mussel or kelp are developed there, and further impact if finfish is developed there because its surface structures are visible from farther away and thus by more people. The response of the benthic environmental health and disease risk sectors is impact (i.e., elevated TOC and risk of outbreak) at a site if the site is developed for finfish aquaculture, and otherwise no impact. Let $R_{n,i,p}$ indicate these responses. We define the maximum response by a sector across all sites and options as $\overline{R}_n \equiv \max_{i,p}\{R_{n,i,p}\}$. Given these definitions, the values are given as follows:

$$V_{n,i,p} = \begin{cases} R_{n,i,p} & \text{if } n = \{\text{aquaculture, halibut}\} \\ \overline{R}_n - R_{n,i,p} & \text{if } n = \{\text{viewshed, benthic health, disease risk}\} \end{cases} \quad (3)$$

The reason why values are calculated differently for these two classes of sectors is that for the aquaculture and halibut sectors, the response is positive, so a higher response increases the value, while for the other sectors, the response is negative, so a higher response decreases the value (so that a higher response indicates less impact).

The final step is to scale the values so they have comparable units. To do so, we scale each sector's value by the domain-wide value that would be attained if the ideal development option to the sector was selected at each site. The result is the scaled value to sector $n$ from applying option $p$ in site $i$, $X_{n,i,p}$, as follows:

$$X_{n,i,p} = \frac{V_{n,i,p}}{\sum_i \max_p (V_{n,i,p})} \quad (4)$$

The scaled values $X_{n,i,p}$ are unitless, range from 0 to 1, and indicate the proportional contribution of development option $p$ at site $i$ to sector $n$'s sum total potential value. This formulation allows us to compare the implications of alternative development options across multiple sectors in multiple sites. The ultimate goal is to select the ideal option at each site. Because society may place different weights on the various sectors, we allow for a weighted value. Let $\alpha_n$ represent the weight placed on sector $n$, so the overall value to sector $n$ at site $i$ from implementing option $p$ is given by $\alpha_n X_{n,i,p}$. Summing over all sectors gives the

comprehensive (all sectors) value from option $p$ at site $i$, as follows:

$$\sum_n \alpha_n X_{n,i,p} \tag{5}$$

Because the goal is to select the option that maximizes this value at site $i$, the maximized value at site $i$ is given by the following objective function:

$$\text{Obj}_i = \max_p \left( \sum_n \alpha_n X_{n,i,p} \right) \tag{6}$$

Evaluation of the above objective function for all $i = 1,\ldots,1061$ aquaculture developable sites generates an MSP solution indicating the optimal location and type (mussel, finfish, kelp, or none) of aquaculture development across the domain, given sector-specific weights $\alpha_n$. Replication of the analysis across a range of weights for each sector generates a set of optimal plans that collectively delineate the 7-D efficiency frontier of MSP solutions (Fig. 2a). In practice, we generated $6^7 = 279,936$ MSP solutions representing six weights $\{\alpha_n = 0, 0.2, 0.4, \ldots 1\}$ applied to each of the seven sectors.

**Conventional planning**. Given the absence of offshore aquaculture development currently in the SCB, and that planning procedures for offshore aquaculture are only in their preliminary stages[28, 69], it is unknown how exactly aquaculture development in the SCB would proceed if directed by a "business as usual", or conventional, approach to planning. In the absence of MSP, a conventional strategy to planning and/or site selection could be driven primarily by the potential economic value to the aquaculture industry. However, this assumes that the industry would be unrestricted in choosing sites for development (apart from assumed fixed constraints like shipping lanes, military areas, MPAs, hard bottom habitat, etc.). Such unrestricted development is unlikely to occur in the SCB due to already established regulations on coastal and offshore development, and regulations for the leasing of state water bottom[28]. In both cases, the State of California mandates that all aquaculture sites demonstrate minimal negative effects on both the environment and existing ocean users (e.g., the existing sectors considered in this study[28]). In this case, conventional planning of aquaculture would not be based solely on potential economic value to the industry. Instead, the aquaculture industry would be expected to consider how to minimize impacts to the environment and existing sectors concurrent with selecting high-value sites for aquaculture. Accordingly, in our model of conventional planning we assumed aquaculture development would focus on sites that have both a high economic value to the aquaculture industry and small negative impacts to existing sectors. To simulate this process, we developed a ratio to determine the suitability of each site for development by each aquaculture sector: the annuity value of the aquaculture sector if the site were developed, divided by the scaled value of the most impacted existing sector at the site if that aquaculture sector were developed there. The rank order of sites in relation to their suitability index was then used to simulate a range of levels of development of aquaculture (1–1061 of the developable sites across the domain) under conventional planning.

We further considered two approaches to conventional planning that reflect variance in the permitting process among the three aquaculture sectors. One approach, "unconstrained conventional planning", promotes free market competition among the three types of aquaculture by allowing the choice of where and what type of aquaculture to develop to be driven solely by the suitability index (compared across sectors and sites). Alternatively, social or political factors may require a more equitable level of development among aquaculture industries. To account for this possibility, we also modeled a "constrained" approach to conventional planning, which regulates for an equal level of development among the three aquaculture farm types. In this case, the first site chosen for development is that with the highest suitability in relation to any type of aquaculture; the second site is that with the highest suitability in relation to the two remaining types of aquaculture; and the third site is that with the highest suitability for the remaining type of aquaculture. The pattern is repeated—maintaining an equal number of sites per sector but also allowing each to choose for development its most suitable sites among those available—up to a set level of development (maximum all 1061 sites; Supplementary Fig. 7a, b). At high levels of development, kelp and finfish aquaculture exhaust their available sites for development; in this case, the selection process focuses on only the sector (s) with available sites.

**Spatial plan outcomes**. Outcomes of the spatial plans derived (under MSP and conventional planning) were calculated for each of the seven sector in terms of their cumulative value achieved across all 1061 developable sites scaled relative to their potential minimum and maximum cumulative values across the sites:

$$O_n = 100 \left( \frac{\sum\limits_i X_{n,i,p} - \sum\limits_i \min\limits_{\forall p}(X_{n,i,p})}{\sum\limits_i \max\limits_{\forall p}(X_{n,i,p}) - \sum\limits_i \min\limits_{\forall p}(X_{n,i,p})} \right) \tag{7}$$

The result is the percentage value of each sector relative to its highest and lowest values possible for the sector. That is, for each aquaculture sector relative to zero value if not developed and the value achieved if it were fully developed across the domain; and for each of the existing sectors relative to its value if maximally impacted by full aquaculture development and its value if not impacted at all due to no aquaculture development. Thus, $O_n$ is scaled relative to the status quo (no aquaculture development) and maximum development/impact. For the halibut sector, which has substantial value outside of the 1061 developable sites (e.g., at the Northern Channel Islands), we considered both developable and non-developable sites in the calculation of its maximum cumulative value in order to not inflate our estimate of the impact of aquaculture on the halibut sector, and thus the halibut fishery's lowest value is ~93%. These outcomes are shown in Fig. 2. We also calculated the NPV and annuity outcomes of the spatial plans for the three aquaculture sectors and halibut sector that could be evaluated in dollars.

**Value of MSP**. The value of MSP relative to a form of conventional planning was the change in outcome to a sector under the two planning approaches being compared, relative to a specified percentage of development for each aquaculture type achieved by both planning approaches. To do this, we first linearly interpolated the 2-D MSP efficiency frontiers in Fig. 2 so that they were continuous functions that could be directly compared to the conventional model outcomes. For each point on a 2-D efficiency frontier we then calculated the difference in value to a sector between the efficiency frontier and the conventional planning outcome with the same level of aquaculture development of each farm type, as illustrated in Fig. 2a. The resulting value of MSP over constrained and unconstrained conventional planning approaches is plotted in Fig. 6. When the same level of aquaculture development does not exist between MSP and a form of conventional planning (e.g., because constrained conventional planning limits mussel aquaculture from reaching full capacity), then the two forms of planning cannot be compared directly, resulting in the incomplete-looking lines in Fig. 6.

**Cluster analysis**. Following Linke et al.[70], we calculated the Bray-Curtis dissimilarity index between each filtered seed plan, represented by an integer vector of the development type at each site. The solution, a matrix of pairwise differences among the plans, was visualized with a hierarchical cluster tree (dendrogram) based on the single linkage algorithm using Euclidean distance and with the number of leaf nodes set to the number of unique spatial plans. Visual inspection identified five clusters, all with multiple spatial plans, representing the complete set of filtered plans. The solution matrix of the Bray-Curtis indices was also visualized in a non-metric multi-dimensional scaling (nMDS) plot using the default Kruskal's Stress 1 criterion. With the aim of covering the maximum amount of variation in spatial design among the filtered set, one seed plan was selected from each cluster based on which one was farthest from the centroid of the nMDS plot.

**Code availability**. Model source code and input data necessary to run the model are available for download at https://github.com/AquacultureSpatialPlanning

**Data availability**. Output data are available on Github (https://github.com/AquacultureSpatialPlanning), and spatial data layers used in our analysis are available on request from the authors.

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

## Acknowledgements

Funding for this project was provided by NOAA Sea Grant (#NA10OAR4170060, California Sea Grant College Program Project #R/AQ-134), The Waitt Foundation, and The Gordon and Betty Moore Foundation. We thank the California Seafloor Mapping Program and CSUMB's Seafloor Mapping Lab for habitat data, S. Denka for assistance running AquaModel simulations, F. O'Brien for AquaModel technical support, T. Ursell for assistance modeling halibut movement, and Catalina Sea Ranch, Santa Barbara Mariculture, Hubbs-SeaWorld Research Institute, and Ocean Approved for information for the aquaculture models.

## Author contributions

S.L., C.W., R.G., S.G., C.C., and L.W. designed research; S.L., C.W., J.S., R.G., C.M., C.K., T.B., and R.S. performed research; S.L., C.W., J.S., R.G., C.K., and T.B. interpreted results; D.K. and J.R. provided analytic tools; S.L., C.W., and J.S. wrote the paper; and all authors contributed to the supporting information.

## Additional information

**Competing interests:** The authors declare no competing financial interests.

