## [Peer Review File · Nature Communications]

Reviewers' comments:

Reviewer #1 (Remarks to the Author):

I invite the authors to revise the manuscript to address specific concerns before a final decision is reached.

Arguments in favour

- The study is sufficiently promising I encourage the authors to resubmit
- Provides evidence for its conclusions
- Of importance to scientists dealing with marine spatial planning and offshore mariculture
- Paper stands out from others in its field by proving a novel methodology
- Can be of interest to researchers in other related disciplines.
- The claims are appropriately discussed in the context of previous literature.
- The manuscript is well written

Arguments against publication

- The methods and results will not be easy to understand by decision-makers managers.
- There is a need to explain how these methods/results would be communicated and disseminated for practical decision-making.
- MSP is a public process, thus the information generated in this study would need to be discussed/validated with relevant stakeholders. Stating that this is a "Full MSP model" is not correct, it is missing the social element.
- Work on stakeholder analysis for MSP and social related aspects for MSP should be included or at least mentioned.
- Better structure the paper to: abstract, methods, results, discussion and conclusions.
- There is some repetition of text, this needs to be avoided as much as possible.
- I estimate about 2-3 weeks for improvements are needed to polish this paper
- Readers could benefit from a simple summary table of main criteria and a schematic diagram of the main methods/results to accompany this paper
- The height of a viewer in the viewshed model is only 1 m tall this needs to be explained.
- Models could be adapted to aquaculture development elsewhere only if the required data is available.
- I did not find the sources in the Annexes for water temperature, currents, etc.
- Need to indicate if experiments could be reproduced elsewhere outside the US.
- Need to indicate where (e.g. in which Web site) the data and code will be made available if this paper is published.

Reviewer #2 (Remarks to the Author):

This is an interesting and well organized paper addressing a complex and important problem. The approach expands substantially on earlier papers addressing similar topics, and the highlighting of aquaculture illustrates well the trade-offs involved in better marine spatial planning.

The only important missing element in the paper is a discussion of the incentives to use a better model. These should be discussed from at least two different perspectives: a) As to some extent is already done in the paper, the potential benefits of a new sector relatively to the costs for existing sectors in an optimal as well as a least impact scenario. b) with the current management system in California, is there any possibility that existing stakeholders can be convinced to allow new uses?

Please check the sentence stating on line 19. It is really hard to read.

Ln 41. Please consider noting that aquaculture has been particularly contentious in the US, and regulations and opposition basically prevent the industry from developing (Knapp and Rubino, Review of Fisheries and Aquaculture, 2015), a feature that could be helped with better planning. Also, because of the economic opportunity, aquaculture is the world's fastest growing food production technology and is expected to continue to grow rapidly (Kobayashi et al, 2015, Aquaculture Economics and Management), and it is expected to continue to fuel increased seafood trade (Asche et al, 2015, World Development), where the US is not the largest importer.

Ln 115. Does there need to be a reduction in benthic quality? Most fish farms I am familiar with are following after harvest and several countries' regulations require a full restoration of the benthic quality before the site can be used again.

More generally, while I understand that a given set of parameters need to be chosen for the simulations, you should note that as technology and knowledge in general improves, influencing the parameters. This is particularly true for finfish aquaculture, which after all is a very young industry. Hence, while useful, your model is conditioned on a given set of knowledge, and it will give different trade-offs and outcomes when run at some point in the future with new parameters.

Ln 196 I would like to see the discussion here expanded. It seems like the incentives for other users are still to block aquaculture when it is costless to them to do so. Hence, even though you have a valuable analysis and you may also use it to find which sites where resistance is likely to be smallest (if it does not also depend on the demographics of those being challenged), if there are no measures in the management system that weights cost and benefits, it may not matter that one can show that society is better off by allowing aquaculture. (In this vein, I really like your discussion of the order of how an aquaculture sector is being rolled out.).

While I can understand (given my ignorance) the relatively short description of the kelp sector, I cannot understand that you do not give a more thorough discussion of finfish. What is the underlying model and growth function? Do your model follow Bjørndal's (1988, Marine Resource Economics) harvest function or is it something else, and if so, how does it differ from the harvest functions in the literature that builds on Bjørndal?

Reviewer #3 (Remarks to the Author):

I found your paper and supplementary material well-written and presented. Since I am neither a fisheries biologist or resource economist, I struggled with the modelling approach and results described in the manuscript. In my practical experience with marine planning so far, I have found that models that can quickly generate hundreds of thousands of optimal spatial plans are rarely used in developing marine plans—from experience with MARXAN in the Great Barrier Reef to various decision support tools used (or usually not used) in developing marine plans in the Americas, Europe, Asia, or Oceania over the past decade. For a variety of reasons, including time, capacity, knowledge and data limitations, among other, marine planning so far has been as much a political-social process as a scientific-technical one.

Marine planners are more interested in finding differences among 3-5 planning alternatives than looking at subtle differences among thousands of "optimal" spatial alternatives. While you can construct objective functions for various sectors for modelling, planners and stakeholders argue for months about the objectives of their plan and only rarely specify objectives that are specific and measurable. I don't see how models that generate the wealth of planning options would contribute anything to facilitate this real-world experience

Clearly a huge gap exists between your analytical approach and the real world of marine planning today. The challenge is to make your science-based framework more approachable to marine planners by explaining it more simply.

Reviewer #1

I invite the authors to revise the manuscript to address specific concerns before a final decision is reached.

Arguments in favour

- The study is sufficiently promising I encourage the authors to resubmit*
- Provides evidence for its conclusions*
- Of importance to scientists dealing with marine spatial planning and offshore mariculture*
- Paper stands out from others in its field by proving a novel methodology*

- Can be of interest to researchers in other related disciplines.*
- The claims are appropriately discussed in the context of previous literature.*
- The manuscript is well written*

Thank you to the reviewer for all of the positive feedback.

Arguments against publication

- The methods and results will not be easy to understand by decision-makers managers.*
- There is a need to explain how these methods/results would be communicated and disseminated for practical decision-making.*

We acknowledge the reviewer's concern that the paper's methods are complex and technical, and thus challenging to access by decisionmakers and managers. To rectify this issue, we now provide a Table (Supplementary Table 1) that outlines the steps for implementing our MSP analytical model framework, their significance, and their application to our case study. We also have better highlighted within the paper ways in which the complex results can be distilled into easy-to-use tools and guidance for spatial planning (e.g., the hotspot maps, outputs of the hypothetical planning exercise). In particular, we added an example in which the large number of optimal spatial plans was refined into a more tractable collection of three seed plans (Fig. 5), as planners and managers often want a small number of plans to compare and adjust based on stakeholder feedback. We also think, however, that methods and results that are intended for use in decisionmaking must first be vetted by technical experts, and that peer reviewed papers like this one are primarily vehicles for communication to this scientific audience. Other types of dissemination materials would be used following publication of this paper to reach decisionmakers, policymakers, and stakeholders (e.g., presentations, user-guide, lay summary). From our collective experience, we have learned that decisionmakers value peer reviewed publications for the rigorous foundation and justification they provide for innovative planning and management approaches. But such peer reviewed publications do not need to function themselves as the how-to guide – other materials and outreach activities can provide more practical guidance for decisionmaking.

- MSP is a public process, thus the information generated in this study would need to be discussed/validated with relevant stakeholders. Stating that this is a “Full MSP model” is not correct, it is missing the social element.*
- Work on stakeholder analysis for MSP and social related aspects for MSP should be included or at least mentioned.*

We defined the shorthand term “full MSP model” as the analytical approach for comprehensive, coordinated and strategic planning. This phrase is not meant to encapsulate the entire MSP process and all of the stakeholder engagement therein. To clarify, we have adjusted the text to refer instead to the “full MSP analytical model,” and have included a statement that a comprehensive marine spatial planning process also includes scientists and managers working with stakeholders to discuss model output, adjust the model based on that feedback, and select a final plan for implementation.

We do not agree that our approach is missing the social element. We are modeling stakeholder concerns related to aquaculture development (viewshed impacts, fishery impacts, etc.). Weighting factors account for differences in stakeholders' preferences across the various planning objectives. These are all important social elements. However, we do agree that discussing and validating model results with stakeholders is an important part of an MSP process, and a tradeoff analysis approach can help frame discussions with stakeholders. We have added text about involving stakeholders in the planning process and how our approach

can integrate into a stakeholder process, and we have referenced previous work about participatory marine spatial planning process (lines 305-321).

•*Better structure the paper to: abstract, methods, results, discussion and conclusions.*

We have followed the structure used in Nature Communication papers.

•*There is some repetition of text, this needs to be avoided as much as possible.*

We have edited the paper with an eye toward removing repetition. However, given that reviewers were also concerned about the accessibility of our paper to diverse audiences, we were reluctant to eliminate the few places where repetition is likely helping to improve clarity. If the reviewer or editor have specific suggestions of repetitive and unnecessary text, we are happy to make additional changes.

•*I estimate about 2-3 weeks for improvements are needed to polish this paper*

•*Readers could benefit from a simple summary table of main criteria and a schematic diagram of the main methods/results to accompany this paper*

We now provide a Table (Supplementary Table 1) that outlines the steps for implementing our MSP analytical model, the significance of each step, and their application to our case study.

•*The height of a viewer in the viewshed model is only 1 m tall this needs to be explained.*

We used the default value in ArcGIS of 1 m, a height which could represent a range of people from children to adults, sitting or standing. However, given the reviewer's comment we have re-run the model at a height of 1.7m, representing an average height of adult men and women in the United States. We have updated all of our results accordingly. Of particular note, because more sites are viewable from land (because of the taller viewer), fewer spatial plans qualified for the filtering criteria used in our hypothetical planning exercise (450 plans rather than ~1300 plans). In updating our model results using the new viewshed model outputs, we also discovered and corrected an error in our halibut fishery model (not all of the fishing sites were previously accounted for in the calculation of the sector's value). As a consequence, unrestricted development of aquaculture has less of an impact on the halibut sector (7%, as opposed to ~22% in our original submission – note that the new percentage still represents > \$100,000 in lost annuity to the halibut fishery). However, our main result is unchanged: strategic development of aquaculture can reduce the impact on the halibut fishery to <1% while achieving significant aquaculture value. We have updated the manuscript text and figures with the correct results.

•*Models could be adapted to aquaculture development elsewhere only if the required data is available.*

Yes, we agree with this statement and have made sure the caveat regarding data availability is clear.

•*I did not find the sources in the Annexes for water temperature, currents, etc.*

In Table S2 (now Supplementary Data 2), we specify that water temperature and current data are derived from the Ocean Circulation Model (OCM). The OCM is described and referenced in Section 3 of the Supplementary Methods.

•*Need to indicate if experiments could be reproduced elsewhere outside the US.*

These models could certainly be adapted, and our analyses reproduced, for another context – U.S. or elsewhere. Our approach and analysis was not specific to a U.S. context, other than using data, concerns/interactions and aquaculture species relevant to California. This is indicated in the last paragraph of the Discussion.

•*Need to indicate where (e.g. in which Web site) the data and code will be made available if this paper is published.*

We have added more specific information to the Data and code availability section of the paper. The model code and necessary input files to run the code are available on Github, the output produced by the model will be made available on Dryad once the manuscript is accepted, and additional spatial data layers used in our analysis are available on request from the authors.

Reviewer #2

This is an interesting and well organized paper addressing a complex and important problem. The approach expands substantially on earlier papers addressing similar topics, and the highlighting of aquaculture illustrates well the trade-offs involved in better marine spatial planning.

Thank you to the reviewer for the positive feedback.

The only important missing element in the paper is a discussion of the incentives to use a better model. These should be discussed from at least two different perspectives: a) As to some extent is already done in the paper, the potential benefits of a new sector relatively to the costs for existing sectors in an optimal as well as a least impact scenario. b) with the current management system in California, is there any possibility that existing stakeholders can be convinced to allow new uses?

We have added more discussion about the incentives to use a better spatial planning model such as ours. A central aim of our paper is to demonstrate an approach that can identify ways to achieve the benefits of the development of a new sector(s) while minimizing or avoiding costs for existing sectors. From this perspective, the incentive to use a better model is that conventional approaches to development decisions in the oceans have often led to high-impact outcomes for existing sectors and for the environment. There is significant momentum for MSP around the world, a clear signal that single sector approaches to planning and management are failing. And often the locations that are the early adopters of MSP are those locations facing impending development of new uses, where managers and stakeholders have recognized that it is more feasible to find a better way to plan for these uses than to try to block them altogether. We explicitly quantify the (positive) value of using a better model over a conventional planning approach (Fig. 6), and in lines 248-259 discuss how this result provides incentives to use a better model for spatial planning.

In the case of California, state regulatory agencies currently are quite strict in permitting new developments in nearshore waters, thereby limiting the incentive for stakeholders to embrace a robust spatial planning model. Nevertheless, the California Department of Fish & Wildlife is working to promote sustainable development of offshore aquaculture in state waters. Given the considerable potential for productive aquaculture in U.S. waters and the economic benefits of development, it seems likely a matter of when and how, rather than if the region will be developed. Thus, in California the incentive for a better model will grow with time. Additionally, there is increasing interest at the federal level (by the Bureau of Ocean Energy

Management) in permitting renewable wind and wave energy development along the California coast (beyond the 3-nautical mile State water border). Although aquaculture and renewable energy are different emerging sectors, simultaneous and comprehensive MSP would be expected to produce positive benefits over conventional planning in either case (lines 243-245; 334-340).

Please check the sentence stating on line 19. It is really hard to read.

We have reworked this sentence to improve clarity.

Ln 41. Please consider noting that aquaculture has been particularly contentious in the US, and regulations and opposition basically prevent the industry from developing (Knapp and Rubino, Review of Fisheries and Aquaculture, 2015), a feature that could be helped with better planning. Also, because of the economic opportunity, aquaculture is the world's fastest growing food production technology and is expected to continue to grow rapidly (Kobayashi et al, 2015, Aquaculture Economics and Management), and it is expected to continue to fuel increased seafood trade (Asche et al, 2015, World Development), where the US is not the largest importer.

We agree with all of these excellent points and have added them along with these supporting citations to the Introduction (lines 40-54).

Ln 115. Does there need to be a reduction in benthic quality? Most fish farms I am familiar with are fallowing after harvest and several countries' regulations require a full restoration of the benthic quality before the site can be used again.

Fallowing of finfish farms is typically used in settings where currents are very slow and/or the farms are arguably too large for the sites. In the United States, farms are typically permitted for multiple years without a stipulation to fallow after harvest or to engage in restoration. Rather, permitting is typically only approved when a proposed farm can demonstrate that the farm is unlikely to have a significant negative impact on the benthos by not being too large for the site (our conservative farm design aligns with this requirement) and only being sited over soft bottom (a requirement of our constraint analysis). Further, permitting is expected only for farms located in water of sufficient depth and with sufficient current speeds to prevent substantial accumulation of detrimental levels of organic material on the seafloor.

For our California case study, our estimates of benthic impacts indicate relative differences among potential farm sites in the organic material flux from the farm to the benthos. Although previous work has demonstrated that in some cases, low levels of additional organic material can actually be beneficial to the benthic community, we made the conservative assumption that all benthic effects are undesirable. We have made changes in the main text to better clarify the meaning of this axis in our tradeoff analysis, including modifying this particular sentence (now lines 128-131), and adding a more detailed discussion of these issues to the Supplementary Methods section 6.3. Importantly, our finfish model could certainly be adjusted to include fallow years for contexts where that is the regulatory standard, which would result in far fewer sites being profitable for finfish aquaculture development. We now state this option in the Supplementary Methods section 5.5.

More generally, while I understand that a given set of parameters need to be chosen for the simulations, you should note that as technology and knowledge in general improves,

influencing the parameters. This is particularly true for finfish aquaculture, which after all is a very young industry. Hence, while useful, your model is conditioned on a given set of knowledge, and it will give different trade-offs and outcomes when run at some point in the future with new parameters.

We agree with this point and that is why we limited our analysis to a 10-year time horizon, as we recognize that the industry is evolving rapidly. One of the most likely predicted effects of improved technology and knowledge is lower costs of operation for all types of aquaculture farms. This would increase the number of profitable developable aquaculture sites in our study domain, potentially enabling a broader range of spatial planning options to minimize environmental impacts and tradeoffs with other uses. We have added this point more explicitly in the main text (lines 369-373) and in the Supporting Methods (Section 5). We would also like to note that this paper is intended as a case study demonstration and proof of concept for our modeling framework (i.e., the full analytical MSP model), and when applied to a real decision-making process, many of the parameters would likely be adjusted for that particular planning situation.

Ln 196 I would like to see the discussion here expanded. It seems like the incentives for other users are still to block aquaculture when it is costless to them to do so. Hence, even though you have a valuable analysis and you may also use it to find which sites where resistance is likely to be smallest (if it does not also depend on the demographics of those being challenged), if there are no measures in the management system that weights cost and benefits, it may not matter that one can show that society is better off by allowing aquaculture. (In this vein, I really like you discussion of the order of how an aquaculture sector is being rolled out.).

While we understand the reviewer's point, we don't completely agree. For one, fighting new developments (particularly in a way that is effective) isn't costless – it takes time, motivation, capacity and financial resources. Stakeholders are willing to invest in those costs to block development when they see significant negative consequences of development. The challenge, of course, is to communicate to stakeholders that, through MSP, negative consequences of development could be substantially less than under conventional planning. In terms of weighing costs and benefits, that is exactly what our tradeoff analysis strives to do: determine development plans that best balance impacts and values. The impacts to the existing sectors and environment are the costs of aquaculture development, while the values of aquaculture are the benefits. Importantly, our analysis considers effects of aquaculture development on each sector, not the aggregate effect on society. Thus, in contrast to the reviewer's comment, we do not simply show that society is better off by allowing aquaculture, and instead provide a means of measuring the management outcome, weighing different costs and benefits of aquaculture development, and providing solutions that most efficiently balance those costs and benefits. In the text referenced by the reviewer, we are highlighting situations where there are not strong tradeoffs and thus balancing costs and benefits is not overly challenging. However, we do agree with the reviewer that ocean users may still work to block aquaculture even when its impacts are low. We think the main issue here is public education. Consequently, we have added further discussion to this paragraph highlighting the need to translate scientific studies such as ours into public education about the potential for minimal impacts from aquaculture development when guided by scientifically-informed planning. We also add text acknowledging that our model does not necessarily capture all of the costs of aquaculture development (lines 220-227).

While I can understand (given my ignorance) the relatively short description of the kelp

sector, I cannot understand that you do not give a more thorough discussion of finfish. What is the underlying model and growth function? Do your model follow Bjørndal's (1988, Marine Resource Economics) harvest function or is it something else, and if so, how does it differ from the harvest functions in the literature that builds on Bjørndal?

The underlying growth function for the finfish model is based on a Von Bertalanffy growth function. We model harvest as occurring every 1.5 years (all adults in the farm are harvested at that time, and then the farm is restocked with juveniles), the typical amount of time that striped bass need to grow to harvest size in our study location. Therefore, we did not follow an optimal harvest function like the one proposed by Bjørndal. We recognize that in reality, farms are likely to harvest whenever it would be most profitable to do so, but this would have been too difficult to implement within the more complex finfish modeling framework, and thus for feasibility and consistency, we held harvesting frequency consistent across our model domain. In order to demonstrate how this type of harvesting decision might be incorporated into this framework, we developed a more complex harvesting decision in the kelp model.

We understand the reviewer's request for more methodological details about the finfish model. In the Supplementary Methods Section 5.5, we reference several reports (we have added more references than were included previously) and a paper that are all publicly available, and we provide more detailed documentation of the model structure – the complexity of the model made it space prohibitive to repeat all of the potentially relevant information described in the references materials. However, we have added some additional details to SI Section 5.5 to provide a more context for the reader.

Reviewer #3

I found your paper and supplementary material well-written and presented. Since I am neither a fisheries biologist or resource economist, I struggled with the modelling approach and results described in the manuscript. In my practical experience with marine planning so far, I have found that models that can quickly generate hundreds of thousands of optimal spatial plans are rarely used in developing marine plans—from experience with MARXAN in the Great Barrier Reef to various decision support tools used (or usually not used) in developing marine plans in the Americas, Europe, Asia, or Oceania over the past decade. For a variety of reasons, including time, capacity, knowledge and data limitations, among other, marine planning so far has been as much a political-social process as a scientific-technical one.

Marine planners are more interested in finding differences among 3-5 planning alternatives than looking at subtle differences among thousands of "optimal" spatial alternatives. While you can construct objective functions for various sectors for modelling, planners and stakeholders argue for months about the objectives of their plan and only rarely specify objectives that are specific and measurable. I don't see how models that generate the wealth of planning options would contribute anything to facilitate this real-world experience

Clearly a huge gap exists between your analytical approach and the real world of marine planning today. The challenge is to make your science-based framework more approachable to marine planners by explaining it more simply.

We appreciate the reviewer's perspective and agree that 1) models that generate a vast number of optimal plans are difficult for planners and stakeholders to digest and thus may not get used, 2) that scientific guidance for marine spatial planning must be able to fit within existing political-social processes to be readily applied, and 3) our science-based framework

requires a certain level of technical scientific expertise to understand, interpret and replicate.

For the first point, we have added an example in which the large number of optimal spatial plans was refined into a set of just three optimal plans to compare and select among (Fig. 5). As stated by the reviewer, shown by Rassweiler (2014), and now discussed in our manuscript (lines 160-171 and 314-319), providing managers and stakeholders with a small subset of optimal plans for negotiation and modification is an effective method for arriving at a final plan that is near-optimal. This technique has advantages over providing scientific guidelines and then having managers and stakeholders design their own plans de novo (Rassweiler et al. 2014). Additionally, as mentioned in our response to some of reviewer #2's comments, we now better highlight ways in which the complex results can be distilled into easy-to-use guidance for spatial planning. For example, the hotspot maps indicate locations generally appropriate for development based on sites most frequently developed across our vast number of optimal plans. Additionally, the filtering approach used in the hypothetical planning exercise (Fig. 4) could be replicated with stricter filtering criteria (e.g., lower levels of allowable impacts to the existing sectors) to generate a smaller solution set of filtered plans. Alternatively, we demonstrated an approach for identifying three seed plans from the filtered set (Fig. 5). Also, planners could specify a narrower range of weighting preferences across objectives; the reason our optimization model identified hundreds of thousands of optimal plans is because we looked across all possible combinations of weighting factors (for heuristic purposes). In practice, selection of a smaller range of weights would focus results on a selection of optimal plans representing a portion of the efficiency frontier. These examples indicate alternative approaches to get from a very large number of optimal plans to a smaller and thus more manageable number of plans for informing negotiation and decision-making.

For the second point, we agree with the importance of effectively integrating scientific guidance with stakeholder engagement in the socio-political process of choosing a final spatial plan. Key to achieving this are generation of a reasonably small number of plans (e.g., the seed plans, discussed above) for stakeholders to negotiate and modify (the reviewer's first point, addressed above), and/or providing general guidance to stakeholders (e.g., via hotspot maps, discussed above) that enable them to develop (near-)optimal plans on their own. We have better highlighted these mechanisms in order to more clearly communicate their utility in assisting real-world planning processes.

Third, in terms of explaining our approach in more simple terms, we have revised the text in various places to improve clarity, as well as provided a Table (Supplementary Table 1) that outlines the steps for implementing our MSP analytical model. However, we would also note that our main goal with this particular publication venue is vetting our approach with experts and providing a framework for assessing tradeoffs when considering multiple emerging uses. Following publication of the peer-reviewed paper, we can develop other dissemination products for non-technical audiences, including a non-technical summary and a user-guide aimed at marine planners. For example, members of our team are currently working with Seasketch to make our data layers and results available for use by stakeholders in a real world participatory planning process in Ventura, California, where the Ventura Shellfish Enterprise seeks to permit 20 100-acre lease blocks for mussel aquaculture. We have modified the manuscript to better highlight how our scientific/technical approach can operate within the realities of political-social planning processes (see paragraph starting on line 305).

Reviewers' comments:

Reviewer #2 (Remarks to the Author):

Thanks for good responses.

Reviewer #3 (Remarks to the Author):

The article is well written and has a good methodology but is overstated and unconvincing in its aim to deliver an analytical approach for MSP. At best, the work presents a comprehensive analytical approach, including trade-off analysis for a specific ocean use, in this case aquaculture. While it does this comprehensively and looks across different types of possible aquaculture in the region as well as its conflicts with a limited amount of broader concerns, it remains in essence a sector-specific approach which is not what MSP is intended to do.

MSP is a public process of analyzing and allocating the spatial and temporal distribution of human activities in marine areas to achieve ecological, environmental and social objectives simultaneously. The authors seem to argue that what is missing is an analytical approach to making trade-offs in a more scientific manner. They are correct in identifying this gap but are not correct in the method delivered to address this problem.

A comprehensive analytical approach that would allow to execute fully the potential MSP as the authors aim to accomplish would have started out from the perspective of the ocean space of the region as a whole and weighed every sector/activity/interest in the exploitation in the region through the lens of the common good + efficiency, effectiveness and equity of the use of the ocean space. It would have made trade-offs along the lines of who benefits from what type of activities, what are long-term gains for the many versus short-term gains for the few. Such approach might have favored for example the development of eco-tourism or other activities in the region and resulted in the decision to locate aquaculture in other areas from what is now considered optimal. Perhaps the aquaculture activities are entirely meant for export meaning the gains of the public space used for aquaculture is reserved mainly for the few (private sector), not the public at large (jobs, income generation, associated activities such as shipping etc.) and might mean that ultimately the socio-economic goals of this ocean space are not met. Addressing such questions is how MSP would fully execute its potential.

MSP is ultimately meant to allow optimal management of a public space (the ocean) so it serves in the most equitable way possible the common interest -- this article developed an analytical approach to optimize aquaculture in a given region; which is different from what fully executed MSP is intended to be

the article's scope and objectives should thus be rewritten to do justice to the real value and contribution this research makes to MSP globally.

Reviewer #4 (Remarks to the Author):

This manuscript is easy to read. The math is appropriate and well done.

My primary concern is that I believe the authors have exaggerated the claim of novelty, maybe we all do. In particular the claims in the abstract: "Currently, we lack an analytical approach for MSP to simultaneously coordinate the development of multiple emerging ocean uses while balancing multiple existing management objectives" and later "Thus, a key gap for MSP science is development and demonstration of an analytical approach for comprehensive, coordinated and strategic planning – the "full MSP analytical model" – and assessment of its value relative to

conventional management," are very odd given there is freely available and widely used software, e.g. Marxan with Zones, that does exactly this. These sentences must be deleted, they are simply false. For the reasons stated here and above I do not see merit in these claims.

Hence my primary concern is that I cannot see the novelty in the paper. The methods are not that new. Finding >250,000 good solutions is not an interesting statement (if we added seven meaningless site to the problem the number of good solutions would multiply by more than 100!). Further, the claim that " we demonstrate substantial value of our framework over conventional planning focused on maximizing individual objectives " is an extremely well known consequence of multi-objective planning.

I was even more surprised by the claim "In particular, rarely are multiple sectors planned or sited concurrent¹⁵, such as coordinated designation of fishery, recreation, aquaculture and shipping areas" when I can think of a dozen such papers (Grantham et al. 2013, Mazor et al. 2014 etc.). I think the authors need to attend to the literature more carefully and dig into what has been happening in MSP over the past decade (let alone the even longer history (decades) of optimised multi-sectoral planning in forestry and on the land in general, where multi-sectoral planning is very common using essentially the same methods and ideas embodied in this paper).

Given the opposition to aquaculture in some parts of the world, the best claim for novelty is around the fact that, in this case, aquaculture appears to be highly compatible with other uses of the sea. The idea that aquaculture can be sited in a way that has minimal impacts on some other industries and the environment is an interesting result that may, or may not, be specific to the case study area. Such a situation has been discovered in other spatial planning trade-off analyses and those papers should also be referenced.

As noted by another referee, generating thousands of "optimal?" plans is not that useful. First, are they all truly optimal? More importantly, people can't assimilate thousands of plan and use them for decisions. Linke et al. and Harris et al. show a clever way of taking lots of very good plans and, using cluster analysis, how to present very good but very different plans. Some more reading about how to present the results of thousands of good plans would assist the ms.

Finally, the idea of evaluating static plans through dynamic process models has already been discussed in the literature (Metcalfe et al 2015).

This paper is better suited to PLoS ONE I think.

Point by point response to reviewer comments for NCOMMS-17-02436

Reviewer comments in **bold**, responses in plain font. Line numbers refer to the version of the manuscript with tracked changes.

REVIEWER #2:

Thanks for good responses.

Thank you for the positive feedback.

REVIEWER #3:

The article is well written and has a good methodology but is overstated and unconvincing in its aim to deliver an analytical approach for MSP. At best, the work presents a comprehensive analytical approach, including trade-off analysis for a specific ocean use, in this case aquaculture. While it does this comprehensively and looks across different types of possible aquaculture in the region as well as its conflicts with a limited amount of broader concerns, it remains in essence a sector-specific approach which is not what MSP is intended to do.

We respectfully disagree that we are providing a sector specific approach, nor do we agree that we are providing a framework that applies only to aquaculture. Models of any of the other sectors, activities and objectives of MSP stated by the reviewer (e.g., eco-tourism, equity of use of ocean space) could be incorporated into our framework and weighted alongside the other objectives in deriving MSP solutions. Furthermore, we applied the framework to the challenge of establishing multiple types of aquaculture (as opposed to just one type) so as to demonstrate how our approach can be used to make siting decisions about multiple uses simultaneously. The methodology is the same if, for example, finfish aquaculture, MPAs, and offshore wind farms are being sited simultaneously through a MSP process. This feature of our framework is stated in lines 334-336, highlighted again in lines 395-399, and then described in the Methods and detailed in the SI (in particular in Section 7: Tradeoff Analysis).

MSP is a public process of analyzing and allocating the spatial and temporal distribution of human activities in marine areas to achieve ecological, environmental and social objectives simultaneously. The authors seem to argue that what is missing is an analytical approach to making trade-offs in a more scientific manner. They are correct in identifying this gap but are not correct in the method delivered to address this problem.

A comprehensive analytical approach that would allow to execute fully the potential MSP as the authors aim to accomplish would have started out from the perspective of the ocean space of the region as a whole and weighed every sector/activity/interest in the exploitation in the region through the lens of the common good + efficiency, effectiveness and equity of the use of the ocean space. It would have made trade-offs along the lines of who benefits from what type of activities, what are long-term gains for the many versus short-term gains for the few. Such approach might have favored for example the development of eco-tourism or other activities in the region and resulted in the decision to locate aquaculture in other areas from what is now considered optimal. Perhaps the aquaculture activities are entirely meant for export meaning the gains of the public space used for aquaculture is reserved mainly for the few (private sector), not the public at large (jobs, income generation, associated activities such as shipping etc.) and might mean that ultimately the socio-economic goals of this ocean space are not met. Addressing such questions is how MSP would fully execute its potential.

MSP is ultimately meant to allow optimal management of a public space (the ocean) so it serves in the most equitable way possible the common interest -- this article developed an analytical approach to optimize aquaculture in a given region; which is different from what fully executed MSP is intended to be. The article's scope and objectives should thus be rewritten to do justice to the real value and contribution this research makes to MSP globally.

In response to the concerns raised by this reviewer in the previous three paragraphs, our framework is flexible enough to consider a broad suite of sectors/activities/interests and to quantify those in terms of values and/or impacts (as appropriate) using a wide variety of metrics (e.g., revenue, numbers of jobs, revenue distribution, seafood production, equity, etc.). With some simplifying assumptions, even short-versus long-term benefits/impacts (to any number of sectors) could be integrated in our framework. Further, our framework weighs every sector's interest, using sector-specific weighting parameters in the MSP objective function. Importantly, the sector interests (objectives) would naturally include, but also extend beyond, sectors seeking specific development rights, like aquaculture. Efficiency of use, overall value through the lens of the common good, and equity in the distribution of the use of the ocean space are sector objectives that could be integrated into and weighted alongside any number of other sector objectives (e.g., maximize profit by industry X; minimize population decline of species Y). These decisions on what to include in the objective function and how to weight them would be case study specific, as would decisions on how many and what new ocean activities to consider allowing in the space. Our framework is infinitely flexible to these decisions.

While no model is perfect, each of the factors raised by the reviewer could be quantified in a model and thus addressed using our framework. Within a single paper, there is not sufficient space to cover every one of the factors raised by the reviewer, so we focused on siting multiple types in the offshore aquaculture activities in relation to a suite of aquaculture and existing sector interests. Our analytical framework, and how to integrate an ocean activity and sector objective into it, is summarized in the Tradeoff analysis section of the Methods, and explained in detail in the SI (Section 7: Tradeoff Analysis). We have now better clarified the Tradeoff analysis section of the Methods (lines 471-495), and in the Discussion we highlighted the flexibility of our approach to other sectors and management objectives (lines 303-311, 334-336 and 395-399).

REVIEWER #4:

This manuscript is easy to read. The math is appropriate and well done.

Thank you, we appreciate this positive feedback.

My primary concern is that I believe the authors have exaggerated the claim of novelty, maybe we all do. In particular the claims in the abstract: "Currently, we lack an analytical approach for MSP to simultaneously coordinate the development of multiple emerging ocean uses while balancing multiple existing management objectives" and later ""Thus, a key gap for MSP science is development and demonstration of an analytical approach for comprehensive, coordinated and strategic planning – the "full MSP analytical model" – and assessment of its value relative to conventional management," are very odd given there is freely available and widely used software, e.g. Marxan with Zones, that does exactly this. These sentences must be deleted, they are simply false. For the reasons stated here and above I do not see merit in these claims.

We disagree that our paper does not provide a novel scientific advance or that we have made false claims. In this same vein, we disagree that Marxan with Zones fully delivers on the need for

comprehensive, coordinated and strategic planning. We greatly appreciate the contributions that Marxan and Marxan with Zones (and other conservation planning software, e.g., C-Plan, Zonation) have made to spatial planning, but we deliberately developed our methodology to go beyond this body of work. Marxan with Zones focuses on minimizing an objective function that combines the sum of sector costs (e.g., to fisheries), representativeness of each planning zone (e.g., habitat in MPAs), and spatial connectivity of planning units. It does not explicitly and solely focus on the actual sector objectives, i.e., maximizing individual sector values (e.g., maximization of fish biomass conserved; maximization of fisheries value), as done by our framework. It also does not weight sector values in the objective function relative to their socioeconomic priorities, as we do. As such, Marxan with Zones does not “...balance multiple objectives”, at least not in the way that we do, solving the spatial planning problem directly in relation to sector values, not other factors.

We believe the difference in opinion to be driven by perspective. The minimum representation constraint within Marxan with Zones’ objective function (Watts et al. 2009) evaluates a spatial plan relative to target levels of particular zone(s) with particular feature(s) set *a priori* by the user (e.g., % habitat in MPAs). Thus, the approach assumes the solution to the spatial planning problem to (approximately) contain a certain amount of ocean space to be allocated to particular zones covering given features. The target levels represent underlying objectives about sector objectives (e.g., total fish biomass in the domain, estimated *a priori* with models), but, importantly, these objectives are not evaluated in the objective function in relation to alternative spatial plans. Thus, Marxan with Zones assumes any spatial plan that meets a set target, whatever its spatial configuration, to achieve the underlying sector objective(s). Simultaneous with meeting the targets, a cost equation in Marxan with Zones’ objective function seeks to minimize impacts to a given set of sectors in the system, as well as connectivity costs of each zone. Thus, some sector objectives are represented implicitly by targets in the representation constraint, while others are represented explicitly in the cost equation, making comparison of sector objectives challenging. Finally, in the cost equation, sector objectives are not weighted uniquely and relative to one another. Thus, Marxan with Zones assumes equivalence in priorities for the alternative sectors in the cost equation; preferences for minimizing costs to some sectors over others – reflecting socioeconomic priorities in the system – are not considered (at least not explicitly) by the objective function.

In contrast with Marxan with Zones, in our analytical framework the objective function seeks to maximize each sector’s objective without restrictions or target levels of each zone (although both approaches limit zones from infeasible sites to keep the analysis at each site relevant and the entire computational process efficient). Thus, our approach does not assume the solution to the spatial planning problem to contain any pre-set design features. Also, each of the sectors in the spatial planning problem are represented independently by their individual objectives, measured in terms of their value achieved/maintained, on a unitless continuous scale, in the same equation in the objective function. Thus, a spatial plan is evaluated explicitly and comprehensively in relation to the sector objectives, which are themselves explicitly comparable. Finally, each sector is weighted in its potential contribution to the objective function. Thus, our approach considers alternative socioeconomic priorities for the sector objectives in the case study, and, because the weights are explicit in the objective function, a spatial plan can be derived for any given set of priorities. When priorities are uncertain, the spatial planning problem can be explored more generally across a range of weights to reveal the nature of the tradeoffs among the sectors, as we did in this paper.

The target levels in Marxan with Zones’ objective function could be set to zero, thereby stripping it of any *a priori* assumption of development level. Further, the sector costs could be quantified as lost value,

making our two frameworks more similar. However, sector objectives that were represented by the target levels would need to be represented – and evaluated in relation to each spatial plan – explicitly in the cost function. Also, weights would need to be introduced to the objective function. Finally, replacement of the simulated annealing heuristic with an analytical function would enable exact optimal planning solutions to be derived. All of this is to say that, while the models could through transformations be made more similar, they are not the same.

Our case study is an excellent example of the utility of our framework. First, there is a negligible amount of offshore aquaculture in southern California and, although it is a highly promising industry, there is no target level of development or even a guarantee that it will be allowed to develop further. Thus, our model appropriately does not aim for a target level of aquaculture development in the study domain. Second, concerns over impact to the environment and existing sectors is a priority in our case study. We find that, despite these concerns, there could be modest levels of aquaculture development that incur minimal impacts on the other sectors. Finally, with both the promise and peril of offshore aquaculture in mind, it is uncertain if its expansion would generate increasingly large impacts on the system and whether there are levels of development beyond which impacts turn severe. We address these questions through evaluation of the spatial planning problem across the range of sector weights, and reveal largely good news on a lack of severe tradeoffs and numerous locations – for each type of aquaculture – that could be developed with high value and minimal impacts. Because all sector objectives – aquaculture and existing sectors – are evaluated explicitly, we are able to provide details about the tradeoffs specific to each sector-by-sector interaction. Thus, the unique aspects of our analysis prove useful in this specific case study, and we expect that there are lots of applications similar to this one that would benefit from our marine spatial planning framework. While we don't see the utility of framing our paper as a point by point response to Marxan with Zones, we hope this detailed explanation helps shed some light on why we claim our approach is novel. We also have made some modifications throughout our manuscript and added more references to previous relevant work (as referenced elsewhere in this response letter) to better clarify its contribution to the spatial planning literature (e.g., lines 42-44, 148-150, 329-334).

Hence my primary concern is that I cannot see the novelty in the paper. The methods are not that new. Finding >250,000 good solutions is not an interesting statement (if we added seven meaningless site to the problem the number of good solutions would multiply by more than 100!).

Unlike Marxan with Zones, which generates numerous near-optimal solutions (“near” because a heuristic is used), our framework determines analytically the exact optimal solution for each parameterization (sector weighting) of the objective function. That is, the numerous solutions our framework generates are not simply “good”, but optimal (clarified on lines 108-109). More to the reviewer's point, whether we generated thousands or hundreds of thousands of optimal solutions is not a key scientific advance of our study; what is important and interesting is that we have an efficient analytical method for explicitly identifying all of these optimal solutions, which is not the case for Marxan with Zones. When a study generates a set of plans for a given parameterization of the objective function using Marxan with Zones, it is because each plan is only an estimate of the true optimal solution. The plans then need to be evaluated as an ensemble to deduce properties about the true solution. In contrast, in our case multiple exact solutions were generated to reveal how optimal marine spatial plans, and their associated sector tradeoffs, shift with changing socioeconomic priorities among the sectors.

As a technical point, the reviewer is incorrect in stating that adding more sites would increase the number of optimal solutions. In fact, the number of optimal solutions is determined by the number of sector-by-weight combinations evaluated in the objective function: $[6 \text{ weights}]^7 = 279,936$ solutions, each specific and optimal to a particular allocation of the six weights among the seven sectors. A solution describes the spatial plan across all planning units.

Further, the claim that “we demonstrate substantial value of our framework over conventional planning focused on maximizing individual objectives” is an extremely well known consequence of multi-objective planning.

We agree that this particular finding is not novel, but supports previous studies indicating the value of marine spatial planning with multiple objectives. In our revision we now reference these studies (lines 392-395). Nonetheless, it is important to demonstrate this value for our particular methodology and case study. Accordingly, we have kept the sentence quoted above by the reviewer but have edited it to state that “we confirm the expectation for substantial value of our framework over conventional planning focused on maximizing individual objectives.”

I was even more surprised by the claim “In particular, rarely are multiple sectors planned or sited concurrently, such as coordinated designation of fishery, recreation, aquaculture and shipping areas” when I can think of a dozen such papers (Grantham et al. 2013, Mazor et al. 2014 etc.). I think the authors need to attend to the literature more carefully and dig into what has been happening in MSP over the past decade (let alone the even longer history (decades) of optimised multi-sectoral planning in forestry and on the land in general, where multi-sectoral planning is very common using essentially the same methods and ideas embodied in this paper).

Thank you for the references; we have edited this sentence for clarity and included them. The term “rare” is meant to be relative. As stated by the Mazor et al. 2014 reference, while the number of examples of siting multiple marine sectors concurrently is increasing, siting of sectors one at a time still dominates. Accordingly, we changed the sentence to read, “In particular, compared with siting sectors one at a time (White et al. 2012), rarely are multiple sectors sited concurrently, such as coordinated designation of fishery, recreation, aquaculture and shipping areas (Watts et al. 2009; Mazor et al. 2014; Grantham et al. 2013)”. Note that the previous sentence in the paragraph sets the context of the discussion in terms of spatial planning in marine systems (we left that sentence unedited). Also, note that examples of previous research siting multiple marine sectors concurrently does not affect the novelty of our study – concurrent siting is only a component to our framework that enhances its utility.

Given the opposition to aquaculture in some parts of the world, the best claim for novelty is around the fact that, in this case, aquaculture appears to be highly compatible with other uses of the sea. The idea that aquaculture can be sited in a way that has minimal impacts on some other industries and the environment is an interesting result that may, or may not, be specific to the case study area. Such a situation has been discovered in other spatial planning trade-off analyses and those papers should also be referenced.

We agree with the reviewer that an important and novel result of our paper is the high compatibility between aquaculture and a suite of other ocean objectives, at least in our study system. We have highlighted this result in the paper (lines 114-130), including in the Discussion (lines 241-243). The reviewer suggests that other spatial planning tradeoff analyses have found similar results. We are unaware of such studies. To our knowledge, previous work on spatial planning for offshore aquaculture

has focused on identifying suitable sites for aquaculture or accounting for potential impacts; exploring interactions and synergies between offshore aquaculture development and other economic uses of the ocean (e.g., wind farms, oil platforms, nearshore aquaculture); or examining tradeoffs with another single objective. None of this work has used a multi-objective tradeoff analysis and optimization to inform siting decisions, which is consistent with the first sentence of this reviewer's comment about a novel component of our study. We have added two sentences to the discussion referencing the existing aquaculture literature that examines site selection issues and co-location opportunities with other uses, and the contribution our paper makes relative to this existing work (lines 268-277).

As noted by another referee, generating thousands of "optimal?" plans is not that useful. First, are they all truly optimal?

Yes, they are optimal solutions that were derived analytically given the sector values and sector weights specified in the objective function. If an important sector is not included, then they will not be optimal, but our framework allows for the addition of more sectors and their objectives. In the broadest sense, the reviewer's critique ("are they truly optimal?") could apply to any optimization model that falls short of considering the infinite number of objectives in a system.

More importantly, people can't assimilate thousands of plan and use them for decisions. Linke et al. and Harris et al. show a clever way of taking lots of very good plans and, using cluster analysis, how to present very good but very different plans. Some more reading about how to present the results of thousands of good plans would assist the ms.

We appreciate the reviewer directing us toward these papers, and have changed the way that we identify seed plans (Fig. 5) to use cluster analysis to identify optimal spatial plans that are most distinct from one another in spatial design. This does not affect any of our conclusions. We have updated our manuscript's text (lines 177-180 and 496-507) and Figure 5. Note, both the filtered plans (Fig. 4) and seed plans (Fig. 5) contain only optimal plans, not just "good" plans.

Additionally, our manuscript highlights other ways in which our complex results can be distilled into easy-to-use guidance for spatial planning. For example, the hotspot maps indicate locations generally appropriate for development based on sites most frequently developed across our vast number of optimal plans, and the filtering approach used in the hypothetical planning exercise (Fig. 4) could be replicated with stricter filtering criteria (e.g., lower levels of allowable impacts to the existing sectors) to generate a smaller solution set of filtered plans.

Finally, the idea of evaluating static plans through dynamic process models has already been discussed in the literature (Metcalf et al 2015).

Thank you for directing us toward this study, which we now reference in our manuscript (lines 317-318). While Metcalfe et al. 2015 do not solve their MSP problem dynamically (they used a static model, Marxan with Zones), they did use a dynamic model to evaluate the value of the solutions they determined using the static model, similar to what we did.

More broadly, to our knowledge no study has achieved dynamic optimization of a terrestrial or marine spatial planning problem as large (>1000 sites), variable (4 planning choices per site) and complex (7 sectors) as the problem we address. The current limitation is overcoming excessive computational run-time. Brown. et al. 2015 (Doi: 10.1890/ES14-00429.1) estimated solutions to two marine spatial planning problems using fully dynamic models, but the problems they addressed were smaller, less variable and less complex.

REVIEWERS' COMMENTS:

Reviewer #4 (Remarks to the Author):

I am reviewer 4 and the authors provided very extensive comments. They have modified their claims of novelty in some places, but not others. I am glad they have included some important references and toned down their claims. I agree with some of their comments and disagree with others.

This is a very good paper but, to be candid, it was unnecessarily arrogant in tone. It is still somewhat overblown in terms of its claims of novelty, although I admit novelty is in the eye of the beholder. It doesn't need to overblow its case to be acceptable or a valuable contribution. Good work will be recognised for what it is.

Point by point response to reviewer comments for NCOMMS-17-02436C

*Reviewer comments in **bold**, author responses in plain font. Line numbers refer to the version of the manuscript with tracked changes (i.e., the version submitted with this document).*

REVIEWER #4:

I am reviewer 4 and the authors provided very extensive comments. They have modified their claims of novelty in some places, but not others. I am glad they have included some important references and toned down their claims. I agree with some of their comments and disagree with others.

This is a very good paper but, to be candid, it was unnecessarily arrogant in tone. It is still somewhat overblown in terms of its claims of novelty, although I admit novelty is in the eye of the beholder. It doesn't need to overblow its case to be acceptable or a valuable contribution. Good work will be recognised for what it is.

We did not mean for our tone to come across as arrogant, nor to be over-stating the novelty of the contribution of our paper. To address these comments, we made a number of small changes that collectively modify the tone of the paper to make a more modest and accurate claim of novelty and importance. These changes include:

- 1) Adding a sentence to the abstract that references the important contributions of previous work, while noting a remaining gap that is addressed by our paper (lines 3-7).
- 2) At first mention of the “full analytical model”, we acknowledge that this is a shorthand used for convenience, as we recognize that the use of “full” was an overstatement of the completeness of our approach (lines 94-95). After that point, we use *‘full’ analytical model* (note term in single quotes) to continue to acknowledge this point and made other small changes to emphasize that our approach was focused on balancing specific objectives (e.g., lines 361-362).
- 3) We deleted unnecessary superlatives and adjectives (e.g., significant, greatly, important, etc.) throughout the paper.